



# Factors controlling pCO₂ variability in the eastern Gulf of Cádiz (SW Iberian Península)

Dolores Jiménez-López[1], Ana Sierra[1], Teodora Ortega[1], Soledad Garrido[2], Nerea Hernández-Puyuelo[1], Ricardo Sánchez-Leal[3], Jesús Forja[1]

[1] Dpto. Química-Física, INMAR, Facultad de Ciencias del Mar y Ambientales, Universidad de Cádiz, Campus Universitario Río San Pedro, 11510 - Puerto Real, Cádiz, Andalucía, España

[2] Instituto Español de Oceanografía. Centro Oceanográfico de Murcia. Varadero 1. E-30740, San Pedro del Pinatar, Murcia, España

[3] Instituto Español de Oceanografía. Centro Oceanográfico de Cádiz. Puerto Pesquero, Muelle de Levante s/n. Apdo. 2609. E-
11006, Cádiz, España

*Correspondence to:* Dolores Jiménez-López (dolores.jimenez@uca.es)

**Abstract**

Spatiotemporal variations of the partial pressure of $CO_2$ ($pCO_2$) were studied during 8 oceanographic cruises conducted between March 2014 and February 2016 in surface waters of the eastern shelf of the Gulf of Cádiz (SW Iberian Península)

between the Guadalquivir River and Cape Trafalgar. $pCO_2$ presented a range of variation between 320.6 and 513.6 μatm, with highest values during summer and autumn and lowest during spring and winter, showing a linear dependence between $pCO_2$ and temperature. The distributions of $pCO_2$ were not homogeneous. Spatially, there was a general decrease from coastal to off-shore stations associated with continental inputs and presented an increase in the zones deeper than 400 m due to the influence of the eastward branch of the Azores Current. On the other side, the study area acted as source of $CO_2$ to the

atmosphere during summer and autumn and as a sink in spring and winter, with a mean value for the study period of -0.18 ± 1.32 mmol m⁻² d⁻¹. In the Guadalquivir and Sancti Petri sections, the $CO_2$ fluxes decreased towards offshore, whereas in the Trafalgar section increased due to the presence of an upwelling. These results highlighted the Gulf of Cádiz as a $CO_2$ sink, with a capture capacity of 14.9 Gg year⁻¹.

## 1. Introduction

Continental shelves play a key role in the global carbon cycle, as this is where the interactions between terrestrial, marine and atmospheric systems take place (Mackenzie et al., 1991; Walsh, 1991; Smith and Hollibaugh, 1993). These zones are considered to be among the most dynamic in biogeochemical terms (Wollast, 1991; Bauer et al., 2013), as they are affected by several factors, particularly high rates of primary production, remineralization and organic carbon burial (Walsh, 1988; Wollast, 1993; de Hass et al., 2002). Continental shelves account for about 10 – 15 % of the ocean primary production and

they contribute approximately 40 % of the ocean's total carbon sequestration, by particulate organic carbon (Muller-Karger et al., 2005).

Generally, waters over the continental shelf account for ~15 % of the global ocean $CO_2$ uptake (-2.6 ± 0.5 Pg C yr⁻¹, Le Quéré et al., 2017). Using direct surface ocean $CO_2$ measurements from the global Surface Ocean $CO_2$ Atlas (SOCAT) database, Laruelle et al. (2014) estimated a sea-air exchange of $CO_2$ in these zones of -0.19 ± 0.05 Pg C yr⁻¹, lower than other studies

published in the last decade (e.g. Borges et al., 2005; Cai et al., 2006; Chen and Borges, 2009; Laruelle et al., 2010; Chen et al., 2013). The discrepancies with respect to this estimation derive from the different definitions of the continental shelf domain and the skewed distribution of local studies (Laruelle et al., 2010). In several studies, it has been observed that the continental





shelves present different behaviours according to their latitude: they tend to act as a sink of carbon (-0.33 Pg C yr$^{-1}$) at high and middle latitudes (30 - 90º) and as a weak source (0.11 Pg C yr$^{-1}$) at low latitudes (0 - 30º) (Cai et al., 2006; Hofmann et

al., 2011; Bauer et al., 2013; Chen et al., 2013; Laruelle et al., 2014, 2017). Laruelle et al. (2010) found differences between the two hemispheres: the continental shelf seas of the Northern Hemisphere are a net sink of $CO_2$ (-0.24 Pg C yr$^{-1}$) and those of the Southern Hemisphere are a weak source of $CO_2$ (0.03 Pg C yr$^{-1}$).

The behaviour of the continental shelf presents a high spatiotemporal variability of the air-sea $CO_2$ fluxes due to various processes, particularly thermodynamic effects, biological processes, gas exchange, upwelling zones and continental inputs

(e.g. Chen and Borges, 2009; Ito et al., 2016). Thermodynamic effects are controlled by the inverse relationship between temperature and solubility (0.0423 ºC$^{-1}$, Takahashi et al., 1993), which produces changes in $CO_2$ dissociation. Biological processes can induce $CO_2$ uptake or release, deriving respectively from phytoplankton photosynthesis that decreases the concentration of inorganic carbon, and respiration by plankton and all other organisms (Fennel and Wilkin, 2009). Both, thermodynamic effects and biological processes are associated with the sea-air $CO_2$ exchange by physical and biological pumps

(Volk and Hoffert, 1985). The effects of upwelling systems generate uncertainty (Michaels et al., 2001). Although this process produces a vertical transport that brings up $CO_2$ and remineralized inorganic nutrients from deep seawater (Liu et al., 2010), upwellings are also responsible for high rates of primary production and a reduction of $pCO_2$ under the equilibrium with the atmosphere (e.g. van Geen et al., 2000; Borges and Frankignoulle, 2002; Friederich et al., 2002). Several studies indicate that these systems act as either a source or sink of $CO_2$ depending on their location (Cai et al., 2006; Chen et al., 2013) and the

ocean considered. Upwelling systems at low latitudes act mainly as a source of $CO_2$ but as a sink of $CO_2$ at mid-latitudes (Frankignoulle and Borges, 2001; Feely et al., 2002; Astor et al., 2005; Borges et al., 2005; Friederich et al., 2008; González-Dávila et al., 2009; Santana-Casiano et al., 2009). Upwelling systems in the Pacific and Indian Oceans act as sources of $CO_2$ to the atmosphere, whereas in the Atlantic Ocean they are sinks of atmospheric $CO_2$ (Borges et al., 2006; Laruelle et al., 2010).

The Gulf of Cádiz, located on the south-west of the Iberian Peninsula, is part of the Iberian/Canaries Current System and the

Eastern Boundary Upwelling System (EBUS) (Borges et al., 2006; Relvas et al., 2007; Arístegui et al., 2009; Laruelle et al., 2010). Although it is really a sub-region of this upwelling system; however, it has a seasonal behaviour due to the coastline configuration and the exchange of water masses with the Mediterranean Sea (Arístegui et al., 2009). Finally, the inner shelf because it is more affected by riverine inputs of nutrients and terrestrial carbon (e.g. Gypens et al., 2011; Vandemark et al., 2011) and by human impact (Cohen et al., 1997). The influence of both factors, riverine inputs and human impact, decrease

towards offshore (Walsh, 1991). Several studies have determined that the inner shelf tends to act as a source of $CO_2$ and the outer shelf as a sink (e.g. Rabouille et al., 2001; Cai, 2003; Jiang et al., 2008, 2013; Arruda et al., 2015). The inner platform (depth less than 40 m) also presents greater seasonal variability of temperature than the outer platform, and consequently the effect of temperature on $pCO_2$ will be greater in the inner platform (Chen et al., 2013).

The Gulf of Cádiz is a geographical domain of considerable interest due to its location. In addition to receiving the outflow of

Mediterranean waters through the Strait of Gibraltar, it receives freshwater inputs from several major rivers, i.e. Guadalquivir, Tinto, Odiel and Guadiana. Various studies have been conducted in this area to evaluate the variability of the sea surface $pCO_2$, although they cover smaller areas and shorter duration of time than this work (González-Dávila et al., 2003; Aït-Ameur and Goyet, 2006; Huertas et al., 2006; Ribas-Ribas et al., 2011) or only a specific area like the Strait of Gibraltar (Dafner et al., 2001; Santana-Casiano et al., 2002; de la Paz et al., 2009). All of these studies, however, have determined that this zone

behaves as a global sink of $CO_2$, with seasonal variations mainly induced by the combination of the fluctuations of biomass concentration and temperature.



In the study reported in this paper, the main objective is to evaluate the spatial and seasonal variation of the sea-surface $CO_2$ on the eastern shelf of the Gulf of Cádiz. In addition, we aim to assess the relative contribution of the temperature and non-temperature effects to the total $CO_2$ concentration, and to determine if the area as a whole acts as a sink or a source of $CO_2$ to the atmosphere over time. To do this, we have analysed a surface measurement database of >26000 values of $pCO_2$ during cruises made between 2014 and 2016 and covering a 0.8º x 1.3º area of the Gulf of Cádiz.

## 2. Material and methods

### 2.1. Study area

This study was carried out over the eastern shelf of the Gulf of Cádiz (Fig. 1), which forms a large basin between the southwest of the Iberian Peninsula and the northwest of Africa, where the Atlantic Ocean connects with the Mediterranean Sea through the Strait of Gibraltar. In the Strait of Gibraltar takes place a bilayer flow, with an upper Atlantic layer flowing towards the Mediterranean basin and a deeper outflow of higher-density Mediterranean waters to the Atlantic Ocean (e.g. Armi and Farmer, 1988; Baringer and Price, 1999; Sánchez-Leal et al., 2017). A similar circulation pattern of opposing flows is found in the Gulf of Cádiz where three main water masses are distributed at well-defined depth intervals and areas: the Surface Atlantic Water (SAW), with coastal and atmospheric influence, inflowing at the shallowest depths; the Eastern North Atlantic Water (ENACW), at an intermediate depth, characterised by low salinity; and the Mediterranean Outflow Water (MOW), entering at the deepest level (Criado-Aldeanueva et al., 2006; Bellanco and Sánchez-Leal, 2016).

The Gulf of Cádiz is part of one of the four major EBUS of the world, the North Atlantic upwelling (e.g. Alvarez et al., 2009), that extends from south of Cape Verde (Senegal) to Cape Finisterre (northwest of Spain). For this reason, the Gulf of Cádiz presents characteristics typical of this EBUS: seasonal variability of a winds system favourable to the coastal upwelling (Fiúza et al., 1982), high biological productivity (Navarro and Ruiz, 2006), a system of fronts and zonal currents (García Lafuente and Ruiz, 2007) and a zone of water exchange between the coastal zone and open ocean (Sánchez et al., 2008). However, the fact that the coastline of the study area runs more in a W-E direction than the overall N-S direction common to all the EBUS phenomena, and the bilayer flow through the Strait of Gibraltar, are two factors that complicate the simple EBUS conceptual model (Peliz et al., 2009).

In addition, the surface circulation in the Gulf of Cádiz is characterised by several different processes. These are: first the presence of an anticyclonic water flow towards the east over the shelf edge as far south as the Strait of Gibraltar, known as the Gulf of Cádiz Current (Sánchez and Relvas, 2003; Peliz et al., 2007); second, in the Trafalgar area an upwelling process occurs, produced by tidal interaction with the topography of the zone; and third, the mixing of surface layers induced by the wind (Vargas-Yáñez et al., 2002; Sánchez and Relvas, 2003; Peliz et al., 2009; Sala et al., 2018). In addition, the centre of the Gulf is under the influence of the eastern-end branch of the Azores Current, producing a front subjected to a mesoscale variability (Johnson and Stevens, 2000; García-Lafuente and Ruiz, 2007; Peliz et al., 2007; Sala et al., 2013).

### 2.2. Field sampling and analysis

The database for this study has been obtained following two different sampling strategies. The first one consisted on taking sea surface measurements underway. Meanwhile, the second one acquired the measurement at several discrete surface stations along three transects perpendicular to the coastline: the Guadalquivir transect (GD), the Sancti Petri transect (SP) and the Trafalgar transect (TF) (Fig. 1). Data was recollected during 8 cruises carried out with a seasonal frequency (spring: ST1 and ST5; summer: ST2 and ST6; autumn: ST3 and ST7; winter: ST4 and ST8) during 2014, 2015 and 2016 (precise dates are indicated in Table 1). All the cruises were carried out on board the R/V Ángeles Alvariño, except the one of summer 2015





(ST6) that was carried out on board the R/V Ramón Margalef. The study area is located between 35.4 and 36.7º N and 6.0 and 7.2º W ($52.8 \cdot 10^2$ Km$^2$).

### 2.2.1. Underway measurements

Sea surface temperature (T), sea surface salinity (S) and the $CO_2$ partial pressure ($pCO_2$) were recorded continuously and were averaged with a frequency of 1 min intervals, from the surface seawater supply of the ship (pump inlet at a depth of 5m). T

and S were measured using a SeaBird thermosalinograph (SeaBird 21) with an accuracy of ±0.01 ºC and ±0.003 respectively. The equilibrator design for determining the $pCO_2$ is a combination of a laminar flow system with a bubble type system, similar to that developed by Körtzinger et al. (1996) and described by Padin et al. (2009, 2010).

The surface water $CO_2$ molar fraction ($xCO_2$) and $H_2O$ were determined using a non-dispersive infrared gas analyser (Licor®, LI 6262) that has a minimum accuracy of ±0.3 ppm. It was calibrated daily using two standards: a $CO_2$ free-air for the blank

and a $CO_2$ sub-standard gas of known concentration (413.2 ppm). $CO_2$ concentration of the sub-standard gas was determined from the comparison with standard gases of NOAA with an uncertainty of 0.22 ppm and measured with a Licor 6262 (±1 ppm). The temperature inside the equilibrator was measured continuously by means of a platinum resistance thermometer (PT100 probe, ±0.1 ºC). A pressure transducer (Setra Systems, accurate to 0.05 %) was used to measure the pressure inside the equilibrator.

The $xCO_2$ was converted into $pCO_2$ according to the protocol described in DOE (2007): corrections by water vapour pressure and water surface temperature have been made since the equipment quantifies in dry air and the temperature registered in the equilibrator is different to the T. The temperature difference between the ship's sea inlet and the equilibrator was less than 1.5 ºC.

### 2.2.2. Fixed stations

Discrete surface samples were taken at 5 m depth, using Niskin bottles (10 L) mounted on a rosette-sampler coupled to a SeaBird CTD 911+, to measure pH, chlorophyll-a concentration and nutrients.

The pH was measured by potentiometer in duplicate using 100 mL of seawater with a glass-combined electrode (Metrohm, 905) calibrated on the total pH scale using a TRIS buffer solution (Zeebe and Wolf-Gladrow, 2001). For chlorophyll-a determination, 1 L of seawater was filtered (Whatman, GF/F 0.7 μm) and frozen (-20 °C) until analysis in the lab. Total

chlorophyll-a was extracted with 90 % pure Acetone, and quantified after 24 hours by fluorometry analysis (Hitachi F-2500) (Yentsch and Menzel, 1963). Nutrient samples for analysis of nitrate and phosphate content were filtered through pre-combusted glass-fibre filters (Whatman, GF/F 0.7 μm) and frozen at -20 °C. Analyses were performed in a segmented flow autoanalyzer (Skalar, San Plus) based on classic spectrophotometric methods (Grasshoff et al., 1983).

Moreover, at these stations, Apparent Oxygen Utilization (AOU) was calculated applying the solubility expression proposed

by Weiss (1974) employing dissolved oxygen values registered by the sensor of the rosette (SeaBird 63) that have been checked using Winkler titrations (±0.1 μmol L$^{-1}$).

The accuracies of the determinations obtained are the following: ±0.003 for pH, ±0.1 μg L$^{-1}$ for chlorophyll-a, ±0.10 μmol L$^{-1}$ for nitrate, ±0.02 μmol L$^{-1}$ for phosphate and ±0.1 μmol L$^{-1}$ for dissolved oxygen.

The corresponding data of T, S and $pCO_2$ were obtained by the underway measurements averaging data corresponding to 0.5

mile around the location of the fixed stations.





### 2.3. Temperature and biological effects on pCO$_2$ calculations

To determine the relative importance of the temperature and biological effects on the changes of pCO$_2$ in sea water, we follow the method proposed by Takahashi et al. (2002). To remove the temperature effect from the observed pCO$_2$, the data were

normalized to a constant temperature, the mean in situ T depending on the focus considered, according to Eq. (1).

$$pCO_2 \text{ at } T_{mean} = (pCO_2)_{obs} \cdot \exp[0.0423 \cdot (T_{mean} - T_{obs})] \tag{1}$$

where the subscripts "mean" and "obs" indicate the average and observed T values, respectively.

The effect of temperature changes on pCO$_2$ has been computed by perturbing the mean pCO$_2$ with the difference between the mean and observed temperature. The pCO$_2$ value at a given observed temperature (T$_{obs}$) was calculated based on Eq. (2).

$$pCO_2 \text{ at } T_{obs} = \text{Mean } pCO_2 \cdot \exp[0.0423 \cdot (T_{obs} - T_{mean})] \tag{2}$$

When the temperature effect is removed, the remaining variations in pCO$_2$ are due to the effect of biology, such as net biological utilization of CO$_2$, the vertical and lateral transport and sea-air exchange of CO$_2$.

The biological effect on the surface water pCO$_2$ in a given area, $(\Delta pCO_2)_{bio}$, is represented by the seasonal amplitude of pCO$_2$ values normalized to the mean T, (pCO$_2$ at T$_{mean}$), using Eq. (1):

$$(\Delta pCO_2)_{bio} = (pCO_{2,\ Tmean})_{max} - (pCO_{2,\ Tmean})_{min} \tag{3}$$

The temperature effect of changes on the mean annual pCO$_2$ value, $(\Delta pCO_2)_{temp}$, is represented by the seasonal amplitude of (pCO$_2$ at T$_{obs}$), using Eq.(2):

$$(\Delta pCO_2)_{temp} = (pCO_{2,\ Tobs})_{max} - (pCO_{2,\ Tobs})_{min} \tag{4}$$

The relative importance of the temperature and biology effects can be expressed by the ratio, T/B:

$$T/B = (\Delta pCO_2)_{temp} / (\Delta pCO_2)_{bio} \tag{5}$$

### 2.4. Estimation of CO$_2$ fluxes

Fluxes of CO$_2$ across the sea-air interface were estimated using the relationship:

$$FCO_2 = \alpha \cdot k \cdot (\Delta pCO_2)_{sea-air} \tag{6}$$

where k (cm h$^{-1}$) is the gas transfer velocity; $\alpha$ is the solubility coefficient of CO$_2$ (Weiss, 1974); and $\Delta pCO_2$ is the difference

between the sea and air values of pCO$_2$. The atmospheric pCO$_2$ (pCO$_2$$^{atm}$) values were obtained from the monthly atmospheric data of xCO$_2$ (xCO$_2$$^{atm}$) at the Izaña Atmospheric Station (Spain), (Earth System Research Laboratory; https://www.esrl.noaa.gov/gmd/dv/data/index.php, last access: 9 January 2019). The xCO$_2$$^{atm}$ was converted to pCO$_2$$^{atm}$ as described in DOE (2007).

The gas transfer velocity, *k*, was calculated using the parameterization formulated by Wanninkhof (2014):

$$k = 0.251 \cdot u^2 \ (Sc/660)^{-0.5} \tag{7}$$

where *u* (m s$^{-1}$) is the mean wind speed at 10 m height on each cruise, obtained from the Shipboard Weather Station; Sc is the Schmidt number of CO$_2$ in seawater, and 660 is the Sc in seawater at 20 ºC.




### 2.5. Statistical analysis

Statistical analyses were performed with IBM SPSS Statistics software (Version 20.0. Armonk, NY). The dataset was analysed using one-way analysis of variance test (ANOVA) for analysing significant differences between cruises for discrete and continuous surface data on hydrological and biogeochemical characteristics. The threshold value for statistical significance was taken as $p < 0.05$.

### 3. Results

#### 3.1. Underway variables

Table 1 gives the ranges of variation and the mean and standard deviation of T, S and $pCO_2$ during the 8 sampling cruises. Figure 2 shows the underway distribution of T and $pCO_2$ in the Gulf of Cádiz.

T values were significantly different among all cruises ($p < 0.05$), varying between 14.3 and 23.4 °C. In general, the samplings made during 2014 presented temperatures higher than 2015 and 2016 (Table 1). For the whole period, the averaged values for

both seasons were highest during summer (21.2 ± 1.3 °C) and autumn (21.0 ± 0.8 °C), with the lowest values during spring (15.5 ± 0.5 °C). During winter, T showed an intermediate value of 17.6 ± 0.9 °C. Spatially T tended to increase from coastal to offshore areas, with a difference of ~11.8 °C (Fig. 2A). No substantial differences were found between the three transects studied (GD, SP and TF) in terms of temperature data. The lowest values of T were detected near the Guadalquivir River mouth and Cape Trafalgar (36.19° N, 6.03° W), due to freshwater inputs and the frequent upwelled waters, respectively.

S values remained practically constant throughout the whole study period, although average values varied significantly among the cruises ($p < 0.05$), with values ranging between 34.68 and 37.06. The highest values were recorded during September 2015 (36.44 ± 0.09) and lowest during February 2016 (35.64 ± 0.08) (Table 1). No spatial or seasonal variations were observed. However, during December 2014, in the area of the Guadalquivir River was measured the lowest salinity value (34.68), related with a storm period that led to very heavy freshwater discharges. On the other hand, TF was the area that presented the highest

mean salinity value for the whole study (36.19 ± 0.25).

During our study period, $pCO_2$ values ranged from 320.6 to 513.6 µatm. Highest values were recorded during summer and autumn of 2014 and 2015 (Table 1), with a similar mean value found for both seasons, 411.7 ± 13.3 µatm and 411.3 ± 10.7 µatm, respectively. The lower mean value (390.3 ± 15.2 µatm) was logged during winter and the lowest mean value (383.9 ± 22.1 µatm) during spring. In general, the $pCO_2$ tended to decrease with the distance to the coast (Fig. 2B). $\Delta pCO_2$ values

ranged between -37.5 ± 14.9 µatm during March 2015, and 16.5 ± 9.5 µatm during October 2014. Moreover, an oversaturation relative to the atmosphere was evidenced during spring and winter for both years. In Fig. 2B a sharply variation of $pCO_2$ can be observed at some zones. These coincide with those stations where discrete water samples were taken. This may be due to the daily variation (day/night) presented by $pCO_2$, since the sampling procedure did not take the same amount of time at each station – it varied in function of the depth of the system in each zone. It is possible that during the daytime, the $pCO_2$ values

were higher than at night.

#### 3.2. Discrete surface variables

Table 2 shows the average values and standard deviation of temperature, salinity, pH, AOU, chlorophyll-a, nitrate and phosphate measured at fixed stations along the three transects during the 8 cruises.

The pH presented significant differences among the cruises ($p < 0.05$), with a range of variation from 7.84 to 8.34. Lowest

mean values were found during summer and autumn of 2014 and 2015 (Table 2), coinciding with the highest average values





of $pCO_2$ were recorded (Table 1). The minimum value of pH were found in September 2015 (7.49 ± 0.03) and the maximum in March 2015 (8.09 ± 0.12).

AOU was significantly different between all the cruises ($p < 0.05$), and a clear seasonal variability was not observed. Values measured ranged from -31.9 to 12.3 µmol $L^{-1}$, with the highest values in December 2014 (7.7 ± 2.1 µmol $L^{-1}$) and the lowest

in March 2015 (-19.1 ± 9.4 µmol $L^{-1}$) (Table 2). During spring was registered the lowest mean value for both years (-10.9 ± 11.7 µmol $L^{-1}$); higher mean values were found in summer (- 6.3 ± 6.1 µmol $L^{-1}$) and winter (4.2 ± 6.3 µmol $L^{-1}$). All values were negative except for those of December 2014; that exception may have been due to the exceptional mixing of the water column caused by the storm.

Chlorophyll-a values presented significant differences among the cruises and between the same seasons of each year ($p <$

0.05). This parameter varied from 0.02 to 2.37 µg $L^{-1}$, with the highest mean value measured in March 2015 (0.76 ± 0.55 µg $L^{-1}$), which coincides with the lowest (negative) mean value of AOU (Table 2). The lowest mean value was in June 2014 (0.18 ± 0.14 µg $L^{-1}$). With reference to the seasons of both years, the highest value was in spring (0.72 ± 0.46 µg $L^{-1}$), followed by winter (0.47 ± 0.31 µg $L^{-1}$), autumn (0.28 ± 0.30 µg $L^{-1}$) and the lowest value in summer (0.22 ± 0.26 µg $L^{-1}$). The SP transect presented the mean lowest value of the whole study (0.33 ± 0.31 µg $L^{-1}$), and the TF zone the highest (0.49 ± 0.37 µg $L^{-1}$).

Nitrate concentration did not show significant differences among all the cruises ($p > 0.05$), ranging between 0.00 and 1.93 µmol $L^{-1}$. The highest value was found in December 2014 (1.05 ± 1.96 µmol $L^{-1}$) and the lowest in June 2015 (0.12 ± 0.14 µmol $L^{-1}$) (Table 2). The highest mean value was recorded in winter (0.87 ± 1.70 µmol $L^{-1}$) and the lowest in summer (0.27 ± 0.45 µmol $L^{-1}$) of both years. TF presented the highest mean concentration for the whole study (0.77 ± 0.76 µmol $L^{-1}$).

Phosphate concentration showed significant differences among all the cruises ($p < 0.05$). By season, the highest mean value

was obtained during autumn (0.31 ± 0.45 µmol $L^{-1}$), although the average data in October 2014 (0.09 ± 0.03 µmol $L^{-1}$) was lower than 2015 (0.50 ± 0.55 µmol $L^{-1}$) (Table 2). The minimum mean value was observed during summer (0.11 ± 0.05 µmol $L^{-1}$). GD presented the highest mean value of the whole study (0.28 ± 0.39 µmol $L^{-1}$), and the lower values were found in the TF, and SP transects, with a similar value, 0.15 ± 0.07 µmol $L^{-1}$ and 0.14 ± 0.09 µmol $L^{-1}$, respectively.

### 3.3. Air-sea $CO_2$ exchange

Table 3 summarizes the mean values and standard deviation for atmospheric $pCO_2$, wind speed, gas transfer velocity and the air-sea $CO_2$ fluxes measured in this study.

The mean wind speeds were relatively similar for the whole study period, ranging between 5.5 ± 2.8 m $s^{-1}$ (March 2015) and 7.7 ± 4.2 m $s^{-1}$ (December 2014). The gas transfer velocity varied between 6.9 ± 0.1 cm $h^{-1}$ in March 2015 and 14.4 ± 0.3 cm $h^{-1}$ in June 2015, since it is very sensitive to changes in wind speed (4.6 cm $h^{-1}$ / m $s^{-1}$ at 20 ºC).

There was a clear seasonal variability in the dataset of $CO_2$ fluxes ($p < 0.05$). The study area acted as source of $CO_2$ to the atmosphere during summer and autumn (0.7 ± 0.3 mmol $m^{-2}$ $d^{-1}$ and 1.2 ± 0.4 mmol $m^{-2}$ $d^{-1}$ respectively) and as a sink in spring and winter (-1.3 ± 1.4 mmol $m^{-2}$ $d^{-1}$ and -1.3 ± 0.01 mmol $m^{-2}$ $d^{-1}$ respectively).

### 4. Discussion

#### 4.1. General trends

Numerous research studies have determined that temperature is one of the most important factors that control the variability of $pCO_2$ in the ocean (e.g. Millero, 1995; Bates et al., 2000; Takahashi et al., 2002; Carvalho et al., 2017), as a consequence of the dependence of the solubility of $CO_2$ with the temperature (Weiss, 1974; Woolf et al., 2016). When $pCO_2$ is affected only

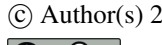



by the temperature, Takahashi et al. (1993) determined a relative variation of $pCO_2$ of 0.0423 ℃$^{-1}$, equivalent to 17.4 µatm ℃$^{-1}$ for experimental $pCO_2$ of 400 µatm.

Figure 3A shows the dependence of the values of $pCO_2$ with T for the entire database, where a linear increase of $pCO_2$ with T ($r^2 = 0.37$, $p < 0.0001$) was observed. This relationship becomes more significant when it is obtained from the mean values of T and $pCO_2$ of each cruise ($r^2 = 0.71$, $p < 0.01$, Fig. 3B). The slope, 4.80 µatm ℃$^{-1}$, is lower than the thermal effect on $pCO_2$ described by Takahashi et al. (1993), and indicates the influence of other processes on the distribution of $pCO_2$ in this zone of the Gulf of Cádiz.

Table 4 gives the values for the dependence of $pCO_2$ with temperature on various continental shelves determined in other studies. The authors of these studies describe the influence of other processes on the relationship between $pCO_2$ and temperature, such as the vertical mixing of the water column (Wang et al., 2000; Xue et al., 2012), continental inputs (Shim et al., 2007; Xue et al., 2012), the influence of surface currents at different temperature (Shim et al., 2007; Jiang et al., 2008), upwelling phenomena (Wang et al., 2005; Jiang et al., 2008; Xue et al., 2012) and biological activity (Wang et al., 2005; Shim

et al., 2007; Ribas-Ribas et al., 2011).

There are previous studies in which the seasonal variations of $pCO_2$ in more coastal zones of the Gulf of Cádiz (depth < 100 m) are described. In 2003, Huertas et al. (2006) found variations of $pCO_2$ ranging between 196 µatm in March and 400 - 650 µatm in August in a zone situated more to the west, between the rivers Guadalquivir and Guadiana. Ribas-Ribas et al. (2011) established during 2006 - 2007, a dependence of $pCO_2$ with temperature similar to that found in this study (5.03 µatm ℃$^{-1}$),

and mean annual values of 369.3 ± 31.3 µatm. Also in 2006, de la Paz et al. (2009) established a variation of $pCO_2$ between 329 µatm in March and 387 µatm in September in the Strait of Gibraltar, a deeper zone situated at the southeastern limit of the Gulf of Cádiz.

There are other studies not concerned with seasonal differences that have quantified $pCO_2$ in different zones of the Gulf of Cádiz. Santana-Casiano et al. (2002) found $pCO_2$ values of 332.2 ± 3.9 µatm in the Strait of Gibraltar during September 1997,

and González-Dávila et al. (2003) determined $pCO_2$ values of 330.6 ± 5.6 µatm on a transect at right-angles to the coastline in the zone of the Guadalquivir, carried out in February 1998. Aït-Ameur and Goyet (2006) reported variations of $pCO_2$ ranging between 300 and 450 µatm for an extensive region of the Gulf of Cádiz in 2002.

Comparing the data given in those previous studies with the mean value found in this study (398.9 ± 15.5 µatm), it is evident that there has been an increase of $pCO_2$ in the Gulf of Cádiz during the last decade, even taking into account the uncertainty

associated with the different measurement techniques employed. When we compare this mean value with the value found in the shallower and deeper studied zones of the Gulf of Cádiz by Ribas-Ribas et al. (2011), who used the same methodology, there has been an increase of $pCO_2$ of 37.6 µatm in the last decade. For the period of time between 2006 and 2016, the rate of growth of $pCO_2$ in the surface waters of the Gulf of Cádiz (3.8 µatm year$^{-1}$) exceeds the rate of increase of $pCO_2$ in the atmosphere (2.3 µatm year$^{-1}$ for the last 10 years in Izaña (Earth System Research Laboratory;

https://www.esrl.noaa.gov/gmd/dv/data/index.php, last access: 9 January 2019); this suggests there may have been changes in the continental inputs of nutrients and C related to anthropogenic activity (Rabouille et al., 2001).

Since the cruises were carried out at the beginning of each meteorological season, it is appropriate to analyse how representative is the range of temperatures that has been obtained. Figure 4 shows the mean value of the last 10 years for the maximum and minimum temperatures in the Gulf of Cádiz acquired by a mooring (bottom-mounted at 36.48° N - 6.96° W;

Puertos del Estado; http://www.puertos.es/es-es/oceanografia/Paginas/portus.aspx, last access: 12 July 2018), and the mean values and standard deviations of the 8 cruises are superimposed. It can be observed that the mean values for each cruise are





within the ranges of variation of the typical temperature in the Gulf of Cádiz, and the mean temperature found, 18.8 °C, is similar to the mean value obtained at the mooring (19.2 °C, Fig. 4). If the dependence of $pCO_2$ with temperature is taken to be 4.80 µatm °C$^{-1}$, one would expect that the mean values of $pCO_2$ obtained in this study would be approximately 2 µatm higher.

The database of this study includes the transition from coastal zones, with depths of the order of 15 - 20 m, to distal shelf waters with depths greater than 800 m. Figure 5 shows the mean values of $pCO_2$ and temperature for different intervals of depth of the water column based on the information obtained in the 8 cruises. It can be observed that the highest values of $pCO_2$ (408.3 ± 26.7 µatm) correspond to the coastal zone (< 50 m), and that values decrease down to 100 - 200 m of depth (396.1 ± 23 µatm). In addition, towards open waters (> 600 m) there is a progressive increase of $pCO_2$ and temperature (404.3

± 16.5 µatm and 20.1 ± 2.4 °C respectively).

Several authors have described the influence of the continental inputs on the distribution of $pCO_2$ in surface waters; the coastal zone is usually oversaturated with $CO_2$, whereas the continental shelf as a whole acts as a sink of atmospheric $CO_2$ (e.g. Rabouille et al., 2001; Chen and Borges, 2009).

The principal continental inputs in the northeast zone of the Gulf of Cádiz take place from the estuary of the Guadalquivir and

from the systems associated with the Bay of Cádiz. De la Paz et al. (2007) found values of $pCO_2$ higher than 3000 µatm in the internal part of the estuary of the Guadalquivir, and Ribas-Ribas et al. (2013) established that this estuary acts as an exporter system of C, nutrients and water oversaturated with $CO_2$ to the adjoining coastal zone. The importance of the contributions from the Guadalquivir on the distribution of $pCO_2$ depends on the river's flow rate, as can be appreciated in Fig. 3. In March 2014 (ST1, green)  high values of $pCO_2$ (up to 500 µatm) were observed in the zone close to the mouth, as a consequence of

the river's high flow rate    (between  192.7  and  299.2  m$^3$  s$^{-1}$, Confederación Hidrográfica del Guadalquivir; http://www.chguadalquivir.es/saih/DatosHistoricos.aspx, last access: 19 Juny 2018) while in the spring of 2015 (ST5, dark green) the lowest values of the study were recorded in this zone (as low as 320 µatm) during a period of drought (flow rate 20 m$^3$ s$^{-1}$) and subject to intense biological activity associated with the highest value found of the concentration of chlorophyll-a (2.4 µg L$^{-1}$).

The Bay of Cádiz occupies an area of 38 km$^2$, and receives urban effluents from a population of 640,000 inhabitants. This shallow zone is oversaturated with $CO_2$ (Ribas-Ribas et al., 2011) due largely to the inputs of $CO_2$, organic matter and nutrients that are received from the river Guadalete and the Sancti Petri and River San Pedro tidal creeks  (de la Paz et al., 2008a, b; Burgos et al, 2018).

Another source of $CO_2$ in the coastal zone results from the net production of inorganic carbon derived from the processes of

remineralization of the organic matter in the surface sediments originated from the continuous deposition of organic matter through the water column (de Haas et al., 2002; Jahnke et al., 2005). The intensity of this process decreases in line with the increasing depth of the system, and the influence of the primary production and the continental supplies on the deposition of the particulate organic matter is less (Friedl et al., 1998; Burdige, 2007; Al Azhar et al., 2017). Ferrón et al. (2009) quantified the release from the sediment of DIC related to the processes of oxidation of organic matter in the coastal zone (depth < 50 m)

of the Gulf of Cádiz, between the Guadalquivir and the Bay of Cádiz. These authors found a mean benthic flux of 27 ± 8 mmol C m$^{-2}$ d$^{-1}$ for stations with a mean depth of 23 m. Considering a well-mixed water column, a pH = 8, in the conditions of mean temperature and salinity in the Gulf of Cádiz (18.8 °C and 36.19, respectively) and using the K1 and K2 acidity constants proposed by Lueker et al. (2000) in the total pH scale. This flux of DIC is equivalent to a $CO_2$ flux of 198 ± 80 µmol C m$^{-2}$ d$^{-1}$, which would produce an increase of $pCO_2$ of 0.25 ± 0.10 µatm d$^{-1}$.



Additionally, in the coastal zone (depth between 50 and 100 m) of the Trafalgar section, an almost permanent upwelling system
        is located (Prieto et al., 1999; Vargas-Yáñez et al., 2002); this system could affect the $pCO_2$ values in this part of the Gulf of
        Cádiz.

        There are other upwelling systems located more to the west of the zone studied. One of these is situated between the cape of
        Santa María and the river Guadalquivir and is more sensitive to meteorological forcing, and there is another at Cape San
Vicente that is almost permanently active (Criado-Aldeanueva et al., 2006). This input of colder waters, with greater loading
        of nutrients and higher concentrations of $CO_2$ (e.g. Liu et al., 2010; Xue et al., 2015; González-Dávila et al., 2017) can affect
        the distributions of $pCO_2$ found in the Gulf of Cádiz.

        The progressive increase of T and $pCO_2$ with increasing depth of the system measured below 100 - 200 m (Fig. 5), it is
        associated with the presence of a branch of the Azores Current that introduces warmer waters in the central part of the Gulf of
Cádiz (Gould, 1985; Käse et al., 1985; Johnson and Stevens, 2000). The influence of warmer surface currents and their
        influence on the variability of $pCO_2$ has been observed in other studies, such as that on the influence of the Gulf Stream in the
        south-eastern continental shelf of the United States (Wang et al., 2005; Jiang et al., 2008), and that on the Kuroshio Current in
        the northern East China Sea (Shim et al., 2007).

        Ribas-Ribas et al. (2011) also found a decrease of $pCO_2$ towards the deep zones (down to ≈ 100 m) on the north-eastern shelf
of the Gulf of Cádiz. In general, an oversaturation of $CO_2$ with respect to the atmosphere in shallower zones and the subsequent
        undersaturation in distal waters, has also been described in other systems such as, for example, in the southern part of the
        Yellow Sea (Qu et al., 2014), in the southwestern part of the Atlantic Ocean (Arruda et al., 2015), in the North Sea (Clargo et
        al., 2015), and on the continental shelf of Maranhense (Lefèvre et al., 2017).

### 4.2. Control factors affecting $pCO_2$

In addition to the influence of temperature, the spatiotemporal distribution of $pCO_2$ in surface seawater is affected by the
        biological utilization of $CO_2$, the vertical and lateral transport, the sea-air exchange of $CO_2$ and terrestrial inputs (e.g. Arruda
        et al., 2015; Ito et al., 2016; Xue et al., 2016).

        With the object of investigating the influence of the biological activity on the variations of $pCO_2$, Fig. 6 shows the dependence
        between the mean values of $pCO_2$ at the fixed stations, and the temperature, pH, AOU and the concentration of chlorophyll-a
(n = 126). The thermal effect on $pCO_2$ is more intense when the discrete database ($pCO_2 = 297 + 5.7$ T, $r^2 = 0.48$; $p < 0.0001$)
        is considered in comparison to the effect obtained using the whole database. The variation of $pCO_2$ with pH ($pCO_2 = 1710 -
        162.8$ pH, $r^2 = 0.34$; $p < 0.0001$), AOU ($pCO_2 = 410 + 1.1$ AOU, $r^2 = 0.21$; $p < 0.0001$), and chlorophyll-a ($pCO_2 = 413 - 20.8$
        [Chl-a], $r^2 = 0.14$; $p < 0.0001$) show the influence of the processes of photosynthesis and respiration on the variations of $pCO_2$
        found. For the Gulf of Cádiz, Huertas et al. (2005) found a linear relationship between $pCO_2$ and chlorophyll-a with a slope
similar to that obtained in this study ($pCO_2 = 274 - 19.6$ [Chl-a], $r^2 = 0.32$; $p < 0.0001$; n = 28). Other authors have also
        described the interrelationships existing between $pCO_2$ and chlorophyll-a in other coastal areas (Borges and Frankignoulle,
        1999; Tseng et al., 2011; Zhang et al., 2012; Qin et al., 2014; Litt et al., 2018). Inverse relationships between $pCO_2$ and
        dissolved oxygen as a consequence of the balance between the processes of photosynthesis and respiration have been found in
        other studies (Zhai et al., 2009; de la Paz et al., 2010; Xue et al., 2012; Xue et al., 2016).

In order to identify the overall controls of temperature and biological effects, the T/B ratio has been calculated (Takahashi et
        al., 2002). A T/B ratio greater than 1 implies the dominance of temperature effects over biological processes, on the $pCO_2$
        dynamics. Strictly speaking, the term that picks up the biological effect in the model of Takahashi et al. (2002), encompasses

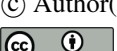



all those processes that are not thermal, that influence the variations of $pCO_2$, including the biological utilization of $CO_2$, continental inputs and the existence of upwellings (Xue et al., 2012; Qu et al., 2014). This method was originally designed for open oceanic systems, but it has been widely used by other authors in coastal areas (e.g. Schiettecatte et al., 2007; Ribas-Ribas et al., 2011; Qu et al., 2014; Burgos et al., 2018).

In this study, the global T/B ratio is 1.15, which indicates that temperature is an important factor controlling intra-annual variation of $pCO_2$. This value is similar to that determined by Ribas-Ribas et al. (2011), in the northeast zone of the shelf of the Gulf of Cádiz, with a ratio of 1.3. De la Paz et al. (2009) propose a T/B ratio of 2.4 in the Strait of Gibraltar, indicating very significant thermal control in this relatively deep zone situated to the east of the Gulf of Cádiz.

Figure 7 presents the values of the T/B ratio grouped in different bottom-depth intervals of the water column in the system. The variations of $\Delta pCO_2$ bio and $\Delta pCO_2$ temp found have been superimposed. In the coastal zone (depth < 50 m), the T/B ratio is below 1 (0.9), and increases to values of 1.3 in the central zone of the Gulf of Cádiz, at depths ranging from 100 to 400 m. However, in the deepest zone (depth > 600 m), a progressive decrease to values of 1.1 is found. This variation of the T/B ratio is largely caused by the variations of $\Delta pCO_2$ bio, with high values close to the coast (120.2 µatm), low values in the central zone (75 µatm) and an increase in the deepest zone (90.7 µatm). Qu et al. (2014) have described the variation on the values of the T/B ratio with the distance to the coast, between 0.4 - 0.6 in the nearshore area (depth < 50 m) to more than 1 (up to 2.4) in the offshore sand (depth > 50 m), in the southern Yellow Sea.

Figure 8 shows the dependence of $\Delta pCO_2$ temp with temperature ($r^2 = 0.40$, $p < 0.01$) and of $\Delta pCO_2$ bio with chlorophyll-a ($r^2 = 0.42$, $p < 0.01$) for the fixed stations; these values confirm the importance of both the thermal and biological processes on the variation of $pCO_2$. The existence of a certain linear correlation between the mean values of chlorophyll-a and the concentrations of nitrate ($r^2 = 0.66$, $p < 0.01$) and phosphate ($r^2 = 0.63$, $p < 0.01$) has also been observed. The increase of the T/B ratio and the decrease of $\Delta pCO_2$ bio from the coastal zone to the central part of the Gulf of Cádiz are associated with the variations of the chlorophyll-a and nutrient concentrations that diminish exponentially with the depth of the system. Thus, the mean concentrations of chlorophyll-a, nitrate and phosphate in the distal zone are 66.3, 81.9 and 44.8 % less than the concentrations found close to the coast.

However, the concentrations of chlorophyll-a and nutrients are relatively constant in waters with bottom-depth to 200 m, and do not explain the decrease of the T/B ratio and the increase of $\Delta pCO_2$ bio in waters with bottom-depth higher than 400 m. These variations have been associated with the change in the origin of the surface water masses. Thus, in the central zone of the Gulf of Cádiz, the origin of the surface waters is a branch of the larger-scale Portuguese-Canary eastern boundary current that circulates around a cyclonic eddy off Cape San Vicente and veers eastward into the Gulf of Cádiz (García-Lafuente et al., 2006). While the deepest zone is under the influence of a branch of the Azores current, which in addition to being a warmer stream that could lead to an increase in primary production, it is the northern border of the subtropical gyre (Klein and Siedler, 1989), thus favour the accumulation of $CO_2$ in this area as convergence zone (Ríos et al., 2005).

The T/B ratios have also been calculated for the different transects at right angles to the coast that have been cruised for sampling in the study zone, as shown in Fig. 9. It can be appreciated that the T/B ratio increases with the distance from the coast on the three transects, and that the temperature generally has a greater influence on the distribution of $pCO_2$ than the non-thermal effects. The T/B ratio varies to the east, with values between 1.0 in the zone of the Guadalquivir and 1.4 in Sancti Petri, and an intermediate value of 1.2 in the Trafalgar zone. These variations are related to changes in the biological activity and the presence of coastal upwellings. The Guadalquivir zone receives substantial continental supplies that lead to high relative concentrations of chlorophyll-a and nutrients; these give rise to high values of $\Delta pCO_2$ bio. In particular, coastal waters



near the mouth of the Guadalquivir River present the highest primary production of all waters within the Gulf of Cádiz (Navarro and Ruiz, 2006). The coastal zone close to Cape Trafalgar has been characterized as a region with high autotrophic productivity and biomass associated mainly with the nutrients input due to upwelling waters (e.g. Echevarría et al., 2002; García et al., 2002). The presence of these emerged water masses could be related to the relatively low values of $\Delta pCO_2$ temp found in this zone. In fact, the mean temperature in this section is 18.4 ± 2.3 °C, about 0.5 °C lower than in the other two zones. The Sancti Petri zone is the one that receives a smaller supply of nutrients, and presents the lowest concentrations of chlorophyll-a in this study. The high values of $\Delta pCO_2$ temp in this part of the Gulf of Cádiz are associated with a higher mean temperature (19.0 °C) and a wider range of variation (6.8 °C).

### 4.3. Ocean-atmosphere $CO_2$ exchange

In the Gulf of Cádiz, the flux of $CO_2$ presents a range of variation from -5.6 to 14.2 mmol m$^{-2}$ d$^{-1}$. These values are within the ranges observed by other authors in continental shelf zones of the North Atlantic (Table 5). As can be appreciated in Fig. 10, the fluxes of $CO_2$ presented seasonal and spatial variations during the period studied. The Gulf of Cádiz acts as a source of $CO_2$ to the atmosphere during the months of summer (ST2, ST6) and autumn (ST3, ST7), and as a sink in spring (ST1, ST5) and winter (ST4, ST8). Previous studies conducted in the Gulf of Cádiz are consistent with the behaviour found in this study (González-Dávila et al., 2003; Aït-Ameur and Goyet, 2006; Ribas-Ribas et al., 2011).

As has been observed with $pCO_2$, temperature is one of the principal factors that control the fluxes of $CO_2$. In fact, for each cruise, a linear and positive relationship has been found between the mean values of the $CO_2$ fluxes and T ($r^2 = 0.72$, $p < 0.01$) (Fig. 11). In parallel, there is a linear and negative relationship between the mean values of the $CO_2$ fluxes and the concentration of chlorophyll-a at the discrete stations sampled ($r^2 = 0.74$, $p < 0.01$) (Fig. 11), as a consequence of the biological utilisation of the $CO_2$ (Qin et al., 2014). These relationships have also been found in various studies carried out in zones similar to the area studied (Zhang et al., 2010; Arnone et al., 2017; Carvalho et al., 2017).

The fluxes of $CO_2$ in the Gulf of Cádiz tend to decrease with the distance from the coast (Fig.10). The coastal zone (< 50 m) presents a mean $CO_2$ flux of 0.8 ± 1.8 mmol m$^{-2}$ d$^{-1}$, that reduces progressively to reach a value of -0.3 ± 1.6 mmol m$^{-2}$ d$^{-1}$ in open waters with bottom-depth higher than 600 m. This dependence of $CO_2$ fluxes with distance from the coast has also been described in other systems, such as in the South Atlantic Bight of the United States (Jiang et al., 2008), in the south-western part of the Atlantic Ocean (Arruda et al., 2015), in the Patagonian Sea (Kahl et al., 2017) and on the continental shelf of Maranhense (Lefèvre et al., 2017). This dependence is the consequence of the decrease of influence of the continental supplies on the $CO_2$ fluxes as one moves towards the open sea. Ribas-Ribas et al. (2011) also found that in the Gulf of Cádiz the $CO_2$ fluxes vary with the distance from the coast; the zone close to the estuary of the Guadalquivir and the Bay of Cádiz acts as a source (1.39 mmol m$^{-2}$ d$^{-1}$) and the zone comprising the rest of the shelf acts as a sink (-0.44 mmol m$^{-2}$ d$^{-1}$).

In addition, on both GD and SP transects a decrease of the $CO_2$ flux is found towards the open ocean, due to the continental inputs associated with the estuary of the Guadalquivir and with the Bay of Cádiz, respectively. On the TF transect, in contrast, it was observed that the zone close to the coast acts as a sink of $CO_2$ (-0.4 ± 1.2 mmol m$^{-2}$ d$^{-1}$), and the deeper zone is a weak source of $CO_2$ to the atmosphere (0.3 ± 1.3 mmol m$^{-2}$ d$^{-1}$). This finding can be explained by the presence of an upwelling close to the coast that is likely to be causing an increase of the production (e.g. Hales et al., 2005; Borges et al., 2005). With reference to this, on the TF transect there are significant differences between the mean surface concentrations of chlorophyll-a and nitrate in the coastal zone (0.63 ± 0.43 µg L$^{-1}$ and 1.09 ± 0.77 µmol L$^{-1}$, respectively) and in deeper zones (0.17 ± 0.12 µg L$^{-1}$ and 0.32 ± 0.33 µmol L$^{-1}$, respectively).





The Gulf of Cádiz, during the period of this sampling, acted as a sink of $CO_2$, with a mean rate of $-0.18 \pm 1.32$ mmol $m^{-2}$ $d^{-1}$, that would give rise to an annual flux of $-0.07$ mol C $m^{-2}$ $yr^{-1}$. The findings of previous studies carried out in the Gulf of Cádiz coincide with the behaviour observed in this study (Santana-Casiano et al., 2002; González-Dávila et al., 2003; Huertas et al., 2006; de la Paz et al., 2009; Ribas-Ribas et al., 2011), with the exception of the study by Aït-Ameur and Goyet (2006) in which it was estimated that the Gulf of Cádiz acts as a source of $CO_2$ to the atmosphere, although that study only corresponds to the
summer season.

## 5. Conclusions

The mean value of $pCO_2$ in the eastern part of the Gulf of Cádiz found in this study ($398.9 \pm 15.5$ µatm) indicates that it is undersaturated in $CO_2$ with respect to the atmosphere ($402.1 \pm 3.9$ µatm). The spatiotemporal variation of $pCO_2$ found responds to the influence of different factors that usually affect its distribution in the littoral oceans. In global terms, when the mean
values of the 8 cruises are considered, temperature ($pCO_2 = 302.0 + 5.16$ T, $r^2 = 0.71$, $p < 0.01$) and biological activity ($pCO_2 = 425.0 - 59.15$ [Chl-a], $r^2 = 0.76$, $p < 0.01$) are the two principal factors that explain the temporal variability of $pCO_2$. Over and above these general tendencies, there are spatial variations associated fundamentally with two other processes. Firstly, the effect of the continental supplies is that, in the coastal zone, principally the area close to the mouth of the Guadalquivir, there is a wider dispersion of the values of $pCO_2$; it is in this area where the lowest and highest values have been observed in the
discrete measurements. In this same coastal zone the highest mean values of $pCO_2$ were found - values that diminish progressively in line with increasing distance from the coast, out as far as an approximate depth of some 400 m. Secondly, there is a relative increase of the temperature and $pCO_2$ in the zone furthest from the coast (depth > 400 m) that is the consequence of a change in the origin of the surface water, with the arrival of a warm branch of the Azores current.

The T/B ratio allows the adequately identification of the factors that control the variability of $pCO_2$ in the Gulf of Cádiz. Its
mean value (1.15) suggests that the distribution is principally controlled by the temperature. A decrease of the ratio has been found, related to the existence of non-thermal processes, mainly taking place close to the coast (at depths of 100 m or less). In the proximity of the Guadalquivir estuary the ratio takes a value of 0.93 due to the continental inputs of C and nutrients, and in the zone around the coastal upwelling off Cape Trafalgar the ratio is 1.09. Furthermore, the actual characteristics of the surface water mass that originates under the influence of a branch of the Azores current also produce a decrease of the T/B
ratio in the deeper zone studied (1.05 for depths > 600 m). In contrast, the highest T/B ratio values have been found in the Sancti Petri section, where values of up to 1.54 are obtained for depths greater than 100 m.

The Gulf of Cádiz acts as a sink of $CO_2$, with a mean capacity of capture for the period sampled of 14.9 Gg $year^{-1}$. The $CO_2$ fluxes present seasonal variation: these waters act as a source of $CO_2$ to the atmosphere in summer and autumn and as a sink in winter and spring. The spatiotemporal variability of $CO_2$ is very similar to that found for the distribution of $pCO_2$, with
larger fluxes close to coast. Based on the information available in the zone, there seems to have been a decrease in the capacity for $CO_2$ capture in the zone in recent decades.

### Author contributions

D.J.-L. wrote the manuscript with contributions from A.S., T.O. and J.F.. D.J.-L. and J.F. processed the experimental data. D.J.-L., T.O. and J.F. conceived the original idea. All authors contributed to the collecting the data.

### Competing interests

The authors declare that they have no conflict of interest.





**Acknowledgments**

D. Jiménez-López was financed by the University of Cádiz with a FPI fellowship (FPI-UCA) and A. Sierra was financed by
the Spanish Ministry of Education with a FPU fellowship (FPU2014-04048). The authors gratefully acknowledge the Spanish
Institute of Oceanography (IEO) for giving us the opportunity to participate in the STOCA cruises. We thank the crews of the
R/V's Angeles Alvariño and Ramon Margalef for their assistance during field work. We are also grateful to Drs. X. A. Padin
and F. F. Pérez (IIM-CSIC) for collaboration on the calibration of the sub-standards of $CO_2$. This work was supported by the
Spanish CICYT (Spanish Program for Science and Technology) under contract CTM2014-59244-C3.

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



## Tables

**Table 1: Date, number of measurements (n), range, average values and standard deviation of sea surface temperature (T), sea surface salinity (S) and pCO$_2$ during the 8 cruises undertaken: March 2014 (ST1), June 2014 (ST2), October 2014 (ST3), December 2014 (ST4), March 2015 (ST5), June 2015 (ST6), September 2015 (ST7) and February 2016 (ST8).**

| Cruise | Date | n | T (°C) Range | T (°C) Mean ± SD | S Range | S Mean ± SD | pCO$_2$ (µatm) Range | pCO$_2$ (µatm) Mean ± SD |
|---|---|---|---|---|---|---|---|---|
| ST1 | 28/03 - 01/04, 2014 | 3874 | 14.3 - 16.4 | 15.4 ± 0.6 | 35.57 - 37.06 | 36.11 ± 0.18 | 365.4 - 513.6 | 396.5 ± 19.0 |
| ST2 | 25/06 - 01/07, 2014 | 4118 | 17.0 - 22.9 | 21.1 ± 0.9 | 35.90 - 36.45 | 36.21 ± 0.15 | 368.7 - 459.5 | 412.9 ± 12.6 |
| ST3 | 01/10 - 07/10, 2014 | 4233 | 16.1 - 23.4 | 21.5 ± 1.3 | 35.80 - 36.79 | 36.26 ± 0.22 | 391.6 - 444.5 | 413.5 ± 9.8 |
| ST4 | 10/12 - 16/12, 2014 | 2938 | 15.6 - 19.1 | 18.1 ± 0.7 | 34.68 - 36.72 | 36.36 ± 0.21 | 369.6 - 444.5 | 388.7 ± 12.9 |
| ST5 | 28/03 - 01/04, 2015 | 3180 | 14.6 - 16.9 | 15.6 ± 0.4 | 35.54 - 36.52 | 36.12 ± 0.14 | 320.6 - 416.5 | 368.6 ± 14.9 |
| ST6 | 19/06 - 25/06, 2015 | 3677 | 17.4 - 22.1 | 20.9 ± 0.8 | 35.63 - 36.92 | 36.40 ± 0.08 | 372.1 - 464.1 | 410.3 ± 13.8 |
| ST7 | 15/09 - 18/09, 2015 | 2575 | 17.0 - 21.9 | 20.6 ± 1.1 | 35.02 - 36.79 | 35.64 ± 0.08 | 387.6 - 457.1 | 407.6 ± 11.2 |
| ST8 | 02/02 - 03/02, 2016 | 1812 | 15.1 - 17.5 | 16.8 ± 0.4 | 35.83 - 36.55 | 36.44 ± 0.09 | 346.2 - 442.6 | 392.9 ± 17.9 |




**Table 2: Number of samples (n) and average values and standard deviation of temperature, salinity, pH, apparent oxygen utilization (AOU), chlorophyll, nitrate and phosphate in surface water samples (at depth of 5m) at fixed stations during the 8 cruises: March 2014 (ST1), June 2014 (ST2), October 2014 (ST3), December 2014 (ST4), March 2015 (ST5), June 2015 (ST6), September 2015 (ST7) and February 2016 (ST8).**

| Cruise | n | Temperature (°C) | Salinity | pH | AOU (µmol L$^{-1}$) | Chlorophyll (µg L$^{-1}$)* | Nitrate (µmol L$^{-1}$) | Phosphate (µmol L$^{-1}$) |
|---|---|---|---|---|---|---|---|---|
| ST1 | 18 | 15.3 ± 0.5 | 36.08 ± 0.14 | 8.06 ± 0.03 | -3.6 ± 8.4 | 0.65 ± 0.37 | 0.96 ± 1.01 | 0.14 ± 0.06 |
| ST2 | 16 | 21.0 ± 1.3 | 36.11 ± 0.11 | 7.97 ± 0.03 | -10.3 ± 5.7 | 0.18 ± 0.14 | 0.42 ± 0.60 | 0.12 ± 0.04 |
| ST3 | 17 | 21.7 ± 0.7 | 36.11 ± 0.14 | 7.97 ± 0.06 | -4.6 ± 3.2 | 0.24 ± 0.29 | 0.34 ± 0.27 | 0.09 ± 0.03 |
| ST4 | 17 | 17.7 ± 0.7 | 36.26 ± 0.27 | 8.05 ± 0.05 | 7.7 ± 2.1 | 0.46 ± 0.33 | 1.05 ± 1.96 | 0.23 ± 0.09 |
| ST5 | 16 | 15.5 ± 0.3 | 36.03 ± 0.13 | 8.09 ± 0.12 | -19.1 ± 9.4 | 0.76 ± 0.55 | 0.68 ± 1.17 | 0.17 ± 0.09 |
| ST6 | 16 | 21.1 ± 1.0 | 36.37 ± 0.05 | 8.01 ± 0.03 | -2.4 ± 3.2 | 0.26 ± 0.34 | 0.12 ± 0.14 | 0.10 ± 0.05 |
| ST7 | 17 | 20.6 ± 1.2 | 35.63 ± 0.02 | 7.94 ± 0.03 | -2.6 ± 5.0 | 0.29 ± 0.31 | 0.37 ± 0.50 | 0.50 ± 0.55 |
| ST8 | 6 | 16.8 ± 0.2 | 36.45 ± 0.05 | 8.09 ± 0.05 | -5.1 ± 3.1 | 0.69 ± 0.32 | 0.41 ± 0.31 | 0.14 ± 0.11 |

*González-García et al. (2018).





**Table 3: Mean values and standard deviation of atmospheric $pCO_2$ ($pCO_2$ µatm), wind speed, gas transfer velocity (k)**
**and $CO_2$ fluxes during the 8 cruises: March 2014 (ST1), June 2014 (ST2), October 2014 (ST3), December 2014 (ST4),**
**March 2015 (ST5), June 2015 (ST6), September 2015 (ST7) and February 2016 (ST8).**

| Cruise | $pCO_2$ atm (µatm) | Wind speed (m s$^{-1}$) | k (cm h$^{-1}$) | $CO_2$ fluxes (mmol m$^{-2}$ d$^{-1}$) |
|---|---|---|---|---|
| ST1 | 398.7 ± 1.8 | 7.7 ± 3.4 | 13.4 ± 0.2 | -0.3 ± 2.3 |
| ST2 | 404.5 ± 0.5 | 7.4 ± 3.4 | 14.0 ± 0.3 | 0.9 ± 1.4 |
| ST3 | 397.7 ± 0.6 | 6.7 ± 4.0 | 11.8 ± 0.4 | 1.4 ± 0.8 |
| ST4 | 399.4 ± 2.2 | 7.7 ± 4.2 | 14.3 ± 0.2 | -1.3 ± 1.7 |
| ST5 | 405.5 ± 0.6 | 5.5 ± 2.8 | 6.9 ± 0.1 | -2.3 ± 0.9 |
| ST6 | 406.1 ± 0.8 | 7.5 ± 4.1 | 14.4 ± 0.3 | 0.5 ± 1.5 |
| ST7 | 398.4 ± 0.7 | 7.0 ± 3.2 | 12.3 ± 0.3 | 0.9 ± 1.1 |
| ST8 | 406.4 ± 0.3 | 6.8 ± 3.1 | 10.6 ± 0.1 | -1.3 ± 1.6 |

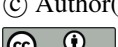



**Table 4: Correlations estimated between pCO$_2$ and temperature (T, ºC) and regression coefficients (r$^2$) in different shelf areas.**

| Site | Correlation pCO$_2$ - T | r$^2$ | Reference |
|------|------------------------|-------|-----------|
| East China Sea | pCO$_2$ = 221 + 5.48 T | 0.9 | Wang et al., 2000 |
| Southeastern continental shelf of the United States | Ln(pCO$_2$) = 5.2505 + 0.0232 T | 0.96 | Wang et al., 2005 |
| Northern East China Sea | pCO$_2$ = 169.7 + 38.19 T | 0.88 | Shim et al., 2007 |
| Southeastern continental shelf of the United States | Ln(pCO$_2$) = 4.611 + 0.058 T | 0.73 | Jiang et al., 2008 |
| Eastern shelf of the Gulf of Cádiz | pCO$_2$ = 269.1 + 5.03 T | 0.42 | Ribas-Ribas et al., 2011 |
| Northern Yellow Sea | pCO$_2$ = 141.3 + 13.7 T (Summer) | 0.56 | Xue et al., 2012 |
|  | pCO$_2$ = 594.5 – 10.7 T (Autumn) | 0.28 |  |
|  | pCO$_2$ = 232.9 + 22.0 T (Winter) | 0.71 |  |
|  | pCO$_2$ = 813.0 – 46.7 T (Spring) | 0.69 |  |
| Continental shelf of Australia | Ln(pCO$_2$) = 4.9 + 0.038 T | 0.74 | Shaw et al., 2014 |
| Gulf of Cádiz | pCO$_2$ = 309.2 + 4.80 T | 0.37 | This work |






**Table 5: Mean average and range of $pCO_2$ and $CO_2$ fluxes (FCO$_2$) in different shelf areas of the North Atlantic.**

| Site | °N | °E | Date | $pCO_2$ (µatm) | $FCO_2$ (mmol m$^{-2}$ d$^{-1}$)* | Reference |
|---|---|---|---|---|---|---|
| North Eastern Atlantic Ocean | 40.0 - 43.0 | -11.0 | May, November, December 1993; July, August 1994 | 320 - 430 | -0.47[a] | Pérez et al. (1999) |
| Upwelling system off the Galician coast | 42.1 - 43.2 | -9.3 – -10.1 | June, July 1997; January, June, July, August, December 1998; January, August, September 1999 | 265 - 415 | -4.7 – -2.3[b] | Borges and Frankignoulle (2002) |
| US Middle Atlantic Bight | 35.0 - 42.0 | -76.0 – -69.0 | February, March, May, June, October 1996 | 220 - 560 | -4.4 – -1.9[b] | DeGranpre et al. (2002) |
| Strait of Gibraltar | 35.6 - 36.0 | -5.5 – -5.2 | September 1997; February 1998; May 1999 | 350 - 354 | -6.9[b] | Santana-Casiano et al. (2002) |
| English Channel | 47.5 - 51.5 | -6.0 – -4.0 | March, September 1995; May, June, July 1997; January, June, July, November 1998; August, September 1999 | 200 - 500 | -20.0 - 12.0[c] | Borges and Frankignoulle (2003) |
| Ría de Vigo | 42.1 - 42.3 | -8.9 – -8.6 | April, July, September, October, November, December 1997 | 285 - 615 | -1.5 - 1.8[a] | Gago et al. (2003) |
| Gulf of Cádiz | 36.3 - 36.7 | -7.0 – -6.5 | February 1998 | 334 - 416 | -19.5 ± 3.5[b] | González-Dávila et al. (2003) |
| Northern and central North Sea | 50.0 - 62.0 | -3.0 – 10.0 | August, September, November 2001; February, March, May 2002 | | 4.6[b] | Thomas et al. (2004) |
| North Sea | 50.0 - 61.0 | -3.0 – 10.0 | August, September 2001 | 220 - 490 | -3.4[b] | Bozec et al. (2005) |
| Gulf of Cádiz | 33.5 - 37.0 | -8.3 – -6.0 | July 2002 | | 18.6 ± 4[b] | Aït-Ameur and Goyet (2006) |
| Northeastern shelf of the Gulf of Cádiz | 36.6 - 37.3 | -7.5 – -6.3 | March 2003 to March 2004 | 196 - 650 | -2.5 - 1.0[b] | Huertas et al. (2006) |
| Southern North Sea | 50.0 - 53.0 | 0.5 – 4.5 | February, March, April, June, August, September, October 2001; February, April, June, August, September, October 2002; August, December 2003; February, May 2004 | 149 - 479 | -23.7 - 6.7[b] | Schiettecatte et al. (2007) |
| US South Atlantic Bight | 28.5 - 34.5 | -81.5 – -76.5 | January, March, July, August, October, December 2005; May 2006 | 330 - 1300 | -3.8 - 3.3[b] | Jiang et al. (2008) |
| Strait of Gibraltar | 35.8 - 36.1 | -6.0 – -5.2 | September 2005; December 2005; March, May 2006 | 320 - 400 | -1.9 - 1.9[b] | de la Paz et al. (2009) |
| Bay of Biscay | 42.0 - 47.5 | -10.0 – -2.0 | September 1994 to December 2004 | 310 - 375 | -11.0 - 0.8[b] | Padin et al. (2009) |
| Gulf of Mexico | 28.0 - 29.3 | -91.0 – -89.7 | August 2004; October 2005; April 2006 | 200 - 600 | -1.17 - 5.4[b] | Lohrenz et al. (2010) |
| Scotian Shelf | 39.5 - 48.0 | -66.0 – -57.0 | July, August, September, October, December 2007; January, February, March, April, May, June 2008 | 203 - 443 | 1.8 ± 1.3[b] | Shadwick et al. (2010) |
| Northern Bay of Biscay | 47.0 - 51.5 | -11.0 – -5.0 | June 2006; May 2007; May 2008 | 248 - 342 | -11.9 – -7.4[d] | Suykens et al. (2010) |
| Northeastern shelf of the Gulf of Cádiz | 36.6 - 37.3 | -6.8 – -6.3 | June 2006; November 2006; February 2007 | 338 - 502 | -2.2 - 3.6[b] | Ribas-Ribas et al. (2011) |
| Portuguese Southern outer continental shelf | 36.0 | -8.0 | October 2001 | 700 - 1130 | 12 ± 8[b] | Oliveira et al. (2012) |
| Northern Gulf of Mexico | 27.5 - 30.5 | -94.0 – -88.0 | August 2004; October 2005; April, June, September 2006; May, August 2007; July 2008; January, April, July, October 2009; March 2010 | 171 - 2222 | -14.3 - 13.1[d] | Huang et al. (2015) |
| Cariaco Basin | 10.0 - 11.3 | -66.3 – -64.0 | March 2004, September 2006; September 2008; March 2009 | 366 - 525 | 0.0 - 10.0[b] | Astor et al. (2017) |
| Mauritanian-Cap Vert upwelling region | 10.0 - 28.0 | -19.0 – -14.0 | From 2005 to 2012 | 275 - 750 | -0.2 - 3.3[c] | González-Dávila et al. (2017) |
| US South Atlantic Bight | 28.0 - 35.0 | -81.0 – -76.0 | July 2007; August, December 2008; May, November 2009; February, April, August, October 2010; March, April, October, December 2011; February, May, August 2012; September 2013; May, July, September, November, December 2014; April, June, July 2015 | 253 - 567 | -1.8 - 2.0[e] | Reimer et al. (2017) |
| Gulf of Cádiz | 35.4 - 36.7 | -6.0 – -7.2 | March, June, October, December 2014; March, June, September 2015; March 2016 | 321 - 514 | -2.3 - 1.5[f] | This work |

*Gas transfer coefficient (k): [a] Woolf and Thorpe (1991), [b] Wanninkhof (1992), [c] Nightingale et al. (2000), [d] Ho et al. (2006), [e] Wanninkhof et al. (2009) and [f] Wanninkhof et al. (2014).

**Figures**

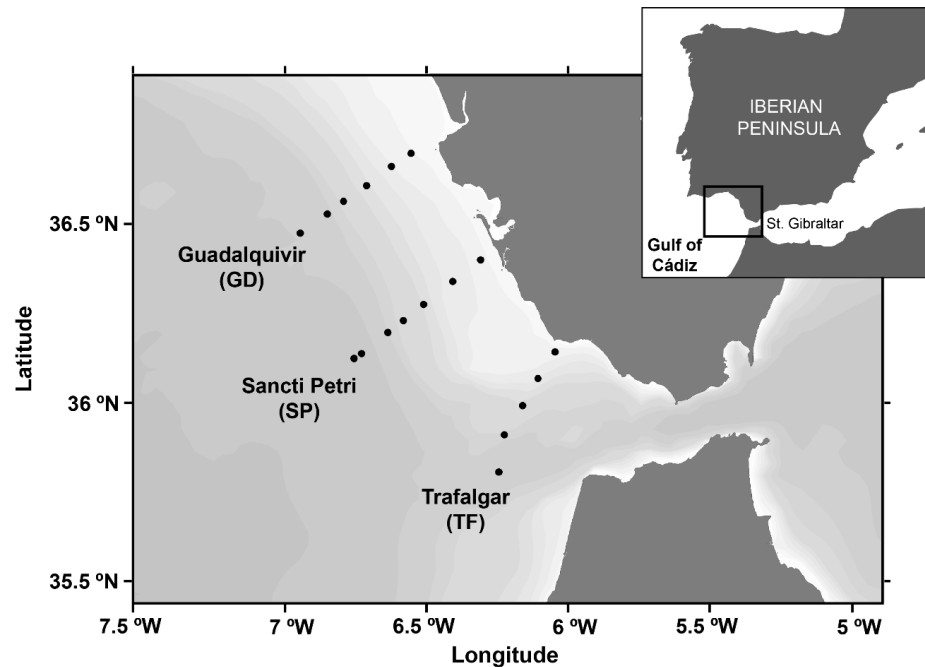

**Figure 1: Map of the eastern shelf of the Gulf of Cádiz showing the location of fixed stations located on 3 transects at right-angles to the coastline: Guadalquivir (GD), Sancti Petri (SP) and Trafalgar (TF).**






**Figure 2: Distribution of sea surface temperature (A) and pCO₂ (B) during the 8 cruises in the Gulf of Cádiz: March 2014 (ST1), June 2014 (ST2), October 2014 (ST3), December 2014 (ST4), March 2015 (ST5), June 2015 (ST6), September 2015 (ST7) and February 2016 (ST8).**






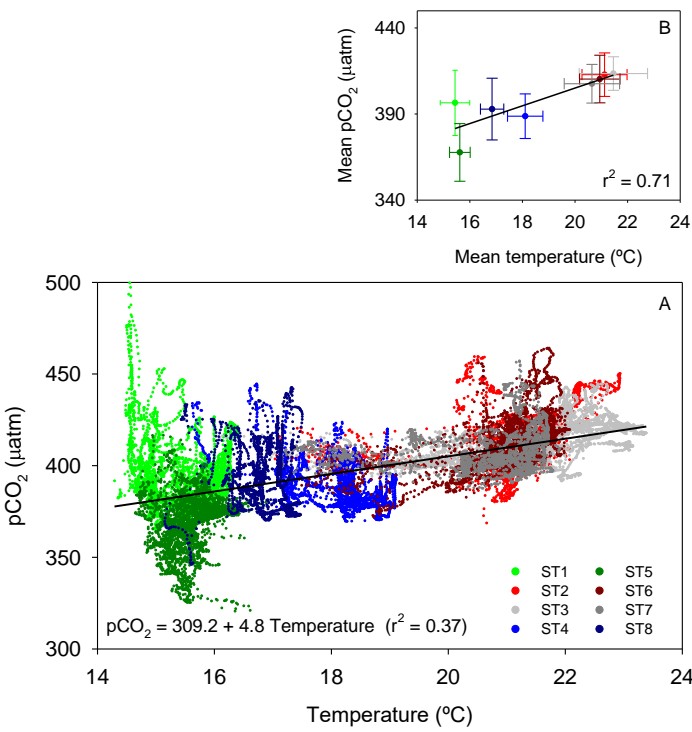

**Figure 3: Dependence of pCO₂ with sea surface temperature for the complete underway database during all the cruises**
**(A) and for the mean values of pCO₂ and temperature for each cruise showing their standard deviations (B). The solid line shows the linear correlation.**




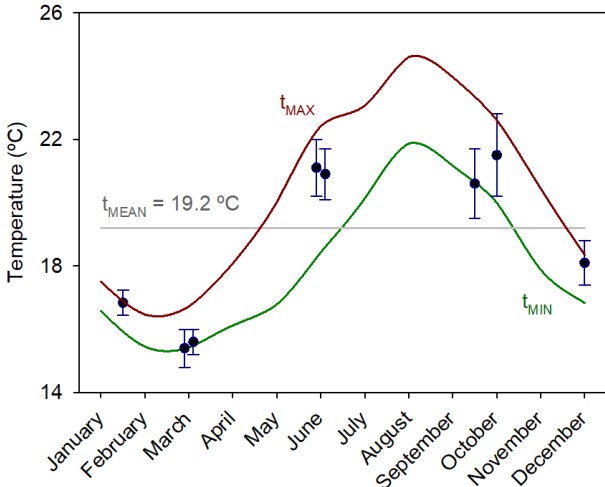

**Figure 4: Maximum and minimum sea surface temperature variation during a 10-year period recorded by a mooring located in the Gulf of Cádiz (36.48°N - 6.96°W). The red line shows maximum sea surface temperature variation. The green line shows minimum sea surface temperature variation. The grey line shows the average temperature for the 10-year period. Blue circles show mean values and standard deviations of sea surface temperature measured during the eight cruises carried out during this study.**






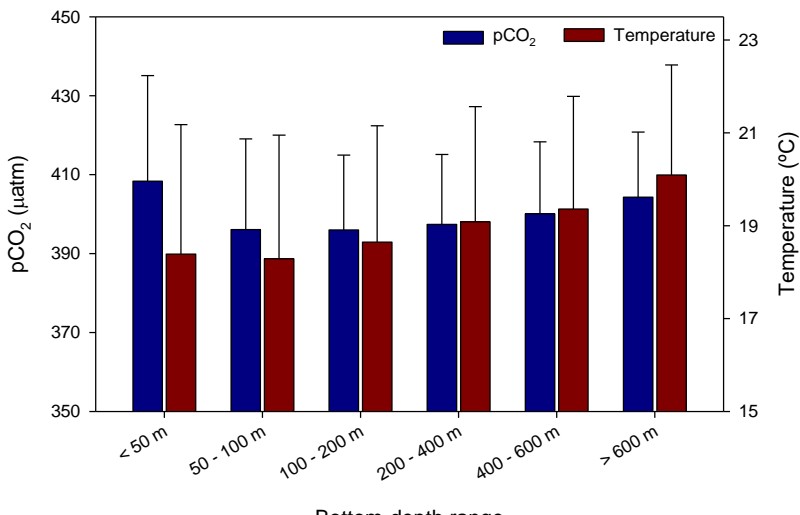

**Figure 5: Variation of $pCO_2$ (µatm) and temperature (ºC) at different bottom-depth ranges of the water column (m) during the 8 cruises. The mean values and standard deviations of $pCO_2$ (blue) and temperature (red) for each range of depth are represented.**





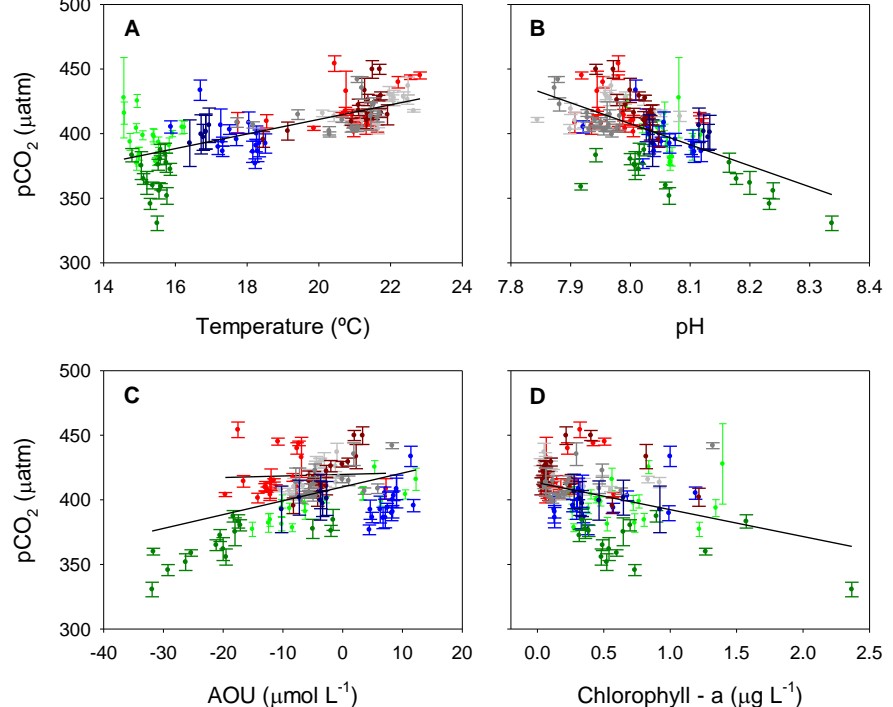

**Figure 6: Dependence between the surface values of pCO₂ and temperature (A, r² = 0.48), pH (B, r² = 0.34), AOU (C, r² = 0.21) and chlorophyll-a (D, r² = 0.14) at the 16 discrete stations during the 8 cruises. pCO₂ presents the standard deviation associated with the mean value obtained from the underway measurements.**






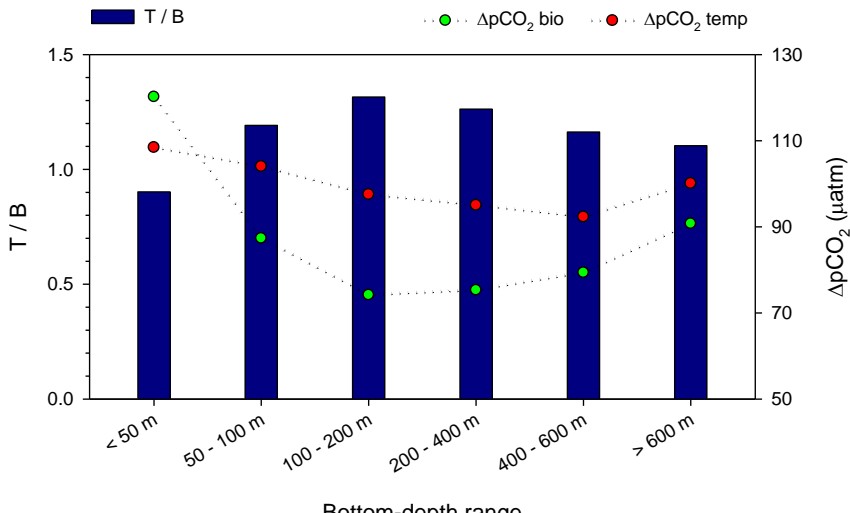

**Figure 7: Variation of the T/B ratio (blue bar), ΔpCO₂ bio (green point) and ΔpCO₂ temp (red point) at different bottom-depth ranges of the water column (m) for the 8 cruises.**





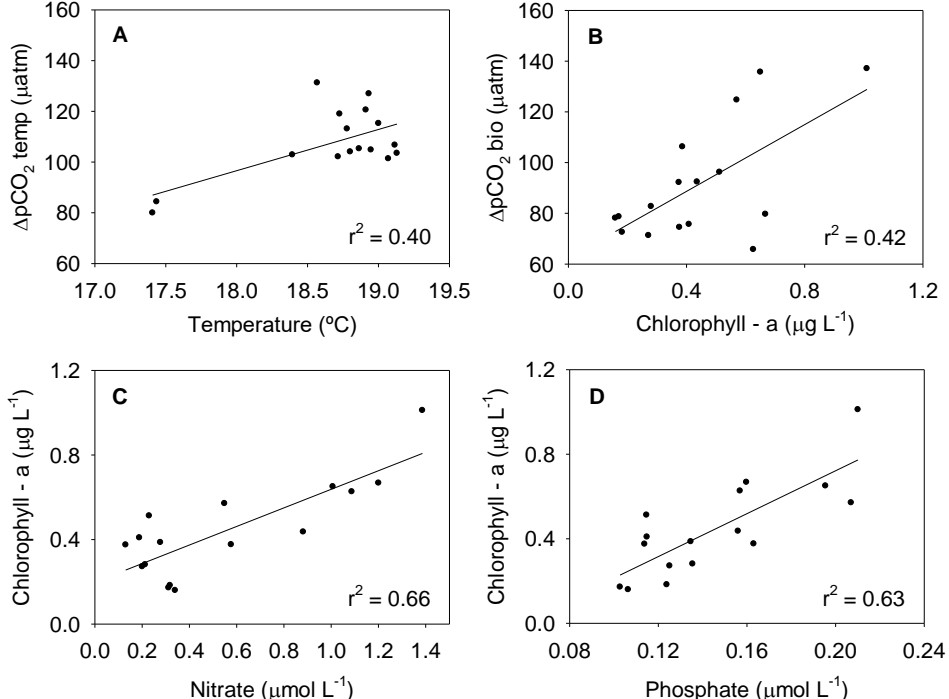

**Figure 8: Correlations between A) ΔpCO₂ temp and temperature, B) ΔpCO₂ bio and chlorophyll-a, C) chlorophyll-a and nitrate and D) chlorophyll-a and phosphate for the mean values at the 16 discrete stations during the 8 cruises.**



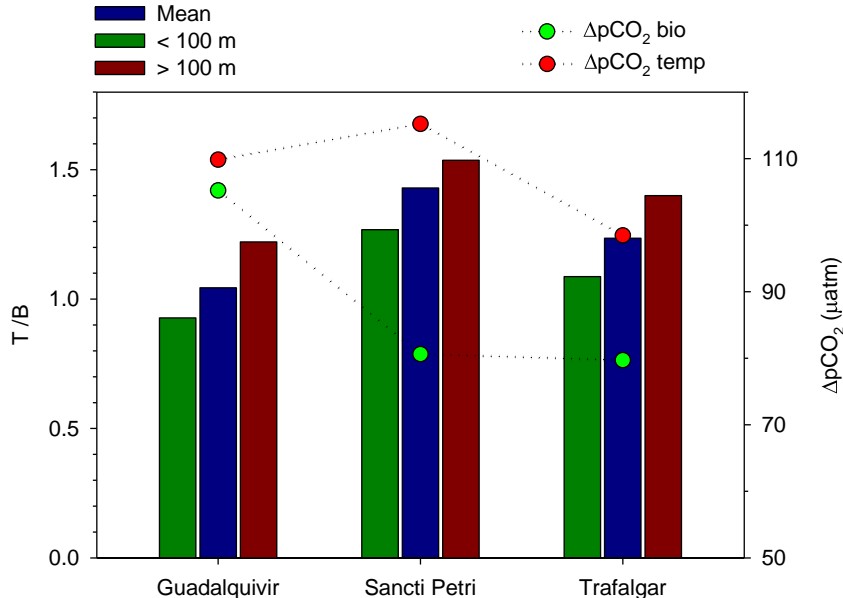


**Figure 9: Variation of the mean T/B ratio (blue bar), the mean T/B ratio at depths < 100 m (green bar), the mean T/B ratio at depths > 100 m (red bar), ΔpCO₂ bio (green point) and ΔpCO₂ temp (red point) on the various transects of the study (Guadalquivir, Sancti Petri and Trafalgar) during the 8 cruises.**







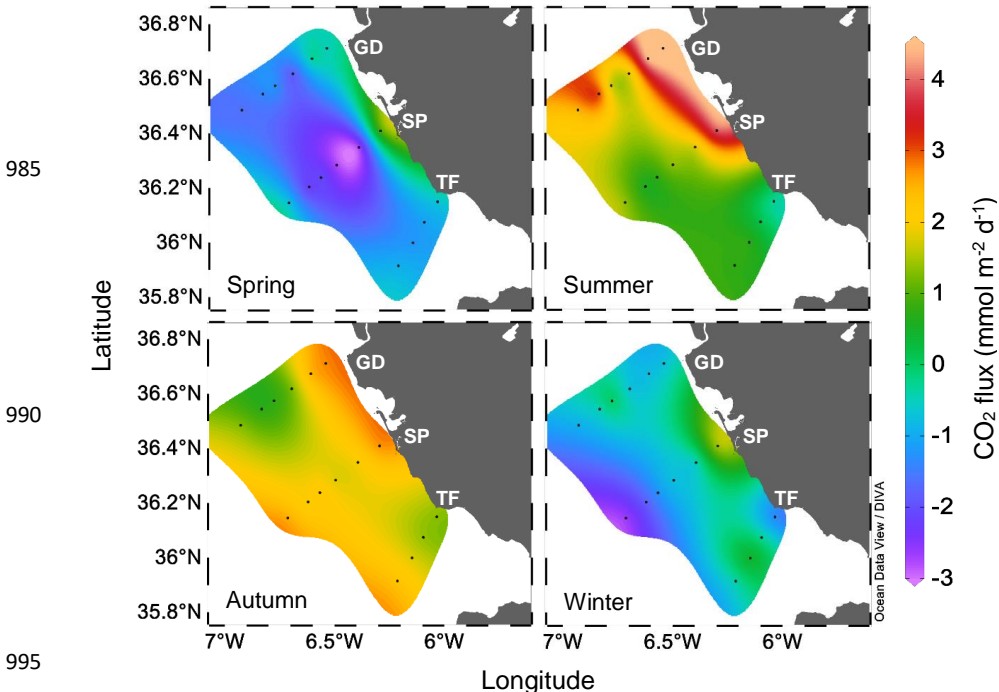

**Figure 10: Spatial distribution of mean values of CO₂ fluxes in the eastern shelf of the Gulf of Cádiz during spring (ST1, ST5), summer (ST2, ST6), autumn (ST3, ST7) and winter (ST4, ST8).**





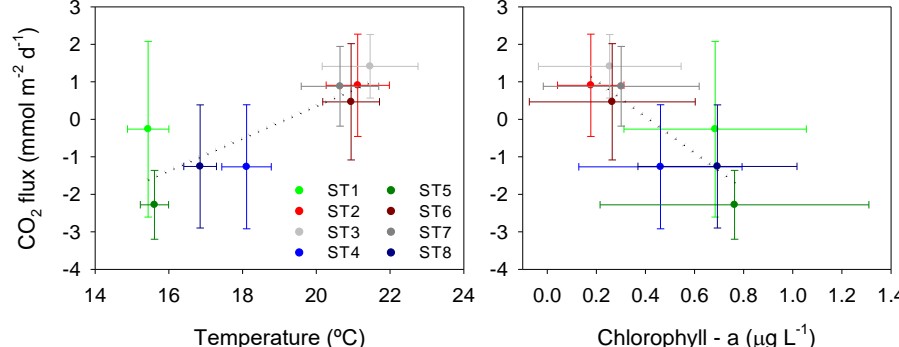


**Figure 11: Correlations between the mean values of CO₂ fluxes and sea surface temperature (T) for the underway database (left, $r^2 = 0.72$), and the CO₂ fluxes and chlorophyll-a at the 16 discrete surface stations (right, $r^2 = 0.74$) for each cruise and showing the standard deviations.**