# Peer review of "pCO2 variability in the surface waters of the eastern Gulf of Cádiz (SW Iberian Peninsula)"

_Ocean Science, 2019_

## Referee Comment (RC1) · Anonymous Referee #1 · 11 Mar 2019

Jiménez-López et al. discuss spatio-temporal variability of $pCO_2$ in the Gulf of Cádiz based on a new dataset collected between March 2014 and February 2016. These results will eventually help to better understand carbon cycle processes on continental shelves and their contribution to the global carbon cycle. And although the authors discuss many different and interesting local features, the submitted manuscript lacks clarity and should be revised and restructured before publication.

General comments

The separation of the driving mechanisms of $pCO_2$ into temperature and biological effects follows the line of argumentation of Takahashi et al. (2002), however, the authors actually calculate thermal and non-thermal components of $pCO_2$ (e.g. Landschützer et al. (2015)). This is also stated by the authors themselves (Line 161 or 372), but

not implemented or followed in their discussion, which is a consequence of the fact that the wording in the method section 2.3 is almost identical to the description of the method in Takahashi et al. (2002) (but not cited as such). Especially in the continental shelves, complex interactions of air-sea gas exchange, primary production, lateral and vertical transport, entrainment of high-DIC waters from below, anthropogenic runoff and freshwater addition lead to changes in salinity, DIC and alkalinity and thereby affect the non-thermal trend in $pCO_2$. Moreover, the authors show seasonality, which they attribute to temperature and biological effects only, while at the same time, they discuss how, e.g., river runoff changes in magnitude over the year and thereby affects $pCO_2$. Although the authors present many different drivers of $pCO_2$ variability, they go back in the temperature-biology framework, which is inconsistent and difficult to follow.

The discussion section needs to be restructured accordingly. First, only results of the 8 cruises should be interpreted without repeating the results. Second, the results should be put in context with previous studies that took place in the same study area and its vicinity; here, it is crucial to include the reference years and seasons. And last, the findings for the Gulf of Cádiz may be compared to other continental shelf areas in the North Atlantic and globally. At the moment, the authors list many results, in the result and discussion section, with little interpretation and no clear line of argumentation that leads to the presented conclusion.

Specific comments

Line 74: If previous studies have already determined the sink strength of the Gulf of Cádiz, seasonally driven by temperature and biology, what is the added value of your study? I am missing a clear motivation for this manuscript in the introduction.

Line 84 / Figure 1: Add bathymetry. Add position of river Guadalete and the tidal creeks and the position of cape San Vicente. You may want to add a circulation scheme here that would help to visualise the surface circulation here.

Line 110: Do the transects cover different water masses or circulation features?

Line 133: How do you correct the temperature difference?

Line 144: How was the oxygen sensor calibrated? Confusing to first explain how AOU is derived, without a detailed description of how oxygen values were determined.

Line 150: Why unit "mile"? Why is exactly 0.5 mile chosen? What is the distance between two stations? Particularly in the SP section, could there be an overlap in $pCO_2$ data for calculating the mean? If there is a CTD coupled to the rosette-sampler, why are not discrete SST and SSS data used for each station and compared / evaluated to the underway SST / SSS measurements?

Line 200: How can there be no spatial and seasonal variation in SSS, when there are are freshwater inputs through storms and rivers?

Line 212: In which zones exactly? If sharp $pCO_2$ variations are observed that coincide with discrete sampling stations, could that be related to the sampling strategy (e.g. potential sampling of ship exhaust) and not be a real signal? Do you correct for this? How can the sampling time be depth-dependent if only discrete samples were taken at 5m depth (Line 135)?

Line 220-228: Are there no spatial differences in pH and AOU?

Line 258: Is it truly equivalent? 17.4 $\mu$atm C$^{-1}$ divided by 400 $\mu$atm results in 0.0435C$^{-1}$.

Line 265: It is not clear to me, why table 4 is useful. Clearly, local effects and seasonality impact $pCO_2$-SST relationships, but they are not discussed or put in perspective with the results of the Gulf of Cádiz.

Line 290: The larger trend in $pCO_2$ in the ocean than in the atmosphere can be driven by

Line 300-305: There is no statistical difference in $pCO_2$ or temperature with bottom depth, which might be because Figure 5 shows data from all seasons and years.

Line 362: You only show the relationship between AOU or pH and $pCO_2$ but there is no discussion of it. Why is almost the entire study area over different seasons oversaturated in oxygen?

Line 377: total or mean T/B. The T/B ratios by Ribas-Ribas et al. (2011) and de la Paz et al. (2009) have been estimated for which years or seasons?

Line 382: How does the DIC flux from the sediment affect T/B?

Line 385: What is the cause for $\Delta pCO_2$bio variations over depth?

Line 389: If $\Delta pCO_2$ temp and bio are calculated as a seasonal amplitude, what temperature and chlorophyll values are used to establish the dependency here? Are these annual means (same for Figure 8 A and B). In any case, I do not understand how the thermal component in relation to temperature and the non-thermal component in relation to chlorophyll confirms the importance of different processes on $pCO_2$ variation.

Line 397: How can the surface chlorophyll and nutrients be constant, when there is a large gradient with distance to the coast (Line 395)?

Line 405: Are the T/B ratios for the different transects significantly different from each other?

Line 422: Again, why is table 5 helpful? I understand that there are many studies that evaluate shelf area processes in the North Atlantic, but this is not discussed in the manuscript. It appears more as a list of literature than it helps to put you own results in perspective.

Figure 3: Panels should have the same size; panel B should next to or below panel A. Add linear correlation equation including units for both panels.

Figure 6: Why are there 2 regression lines plotted for AOA-$pCO_2$?

Figure 7 / 9: What are the uncertainties of the thermal and non-thermal components? Are they significantly different from each other?

Figure 10: You could change the colourbar; it is not clear where the border between outgassing and uptake is located.

You could simply state in section 2.5. that all reported linear correlations are statistical significant with p-values smaller than 0.05 in the entire manuscript unless indicated otherwise. With that, you do not have to report the p-value again. There a numerous linear relationship equations in the manuscript without units. The correlation equations could be plotted within the according figures to increase readability.

Please have colour blind people in mind for all figures. It is not possible to differentiate between years with the presently used lighter / darker colours; you could use different symbols as well.

Abbreviations that are only used in one paragraph only are superfluous; for example EBUS. Consider abbreviating T by SST and and S by SSS for readability.

The manuscript will benefit from the input of a native speaker. There is a need to check for incomplete sentences and the use of correct tenses. There should be fewer, but longer paragraphs that consist of more than one or two sentences; while covering the same topic. This will make it easier to follow clear arguments.

---

## Referee Comment (RC2) · Anonymous Referee #2 · 11 Apr 2019

In general, the manuscript is written with too many very short paragraphs. The topic of the presented research is interesting and pertinent in the actual context of climate change. The title is very appealing and the reader expects to learn about the processes that control the pCO2 variability in the eastern Gulf of Cadiz. Unfortunately, the abstract does not reveal any factor controlling the pCO2 variability. The conclusion indicates that: "temperature and biological activity are the two principal factors that explain the temporal variability of CO2". This is the case everywhere in the ocean. There is nothing new here! Then (line 470), it is indicated that: ". . . the distribution is principally controlled by the temperature". Here again, there is nothing new and this is normal.

Therefore, I would suggest the authors to change the title of their manuscript to more accurately reflect its content. The authors should also precise (and justify) what kind of

linear regression (type I, type II, . . .) they did in order to determine the general trends. They should also detail all their calculations (including simple ones such as the mean capture capacity, . . .). The last sentence should be re-written to me more precise (what means ". . .variability of CO2. . ."; is it TCO2, CO2 flux, pCO2???; what means ". . .capacity of CO2 capture. . ."; surface/depth, on what timescale?). The manuscript will gain in clarity if it were much concise. There is a need to remove some of the tables and figures.

In summary, this manuscript needs a major revision before publishing.

––––––––––––––––––––––––––––––––––––

---

## Referee Comment (RC3) · Anonymous Referee #3 · 24 Apr 2019

The authors investigate factors controlling pCO2 variations in the Gulf of Cadiz. They use high quality data from 8 cruises incorporating underway data of pCO2, SSS, SST, and wind speed as well as discrete data for pH, AOU, and nutrients taken along three repeat transects during the cruises. They present spatiotemporal distributions of the underway data, the cruise averages of the discrete data, and the seasonal changes of the computed air-sea CO2 flux. The authors then discuss the factors influencing the pCO2 variability. Specifically, they quantify thermal/non-thermal controls of pCO2. They conclude that temperature and biological activity are the two principal factors that explain the temporal variability of pCO2. They also point out that continental inputs and mixing with water originating from warm ocean currents influence the spatial variability of pCO2.

[Figure]

The work is OK structured, includes original research based on high quality data, and suits for publication in this journal. However, there are several things that need improvements and/or clarification and I recommend major revision.

General comments: 1- The main subject of the study is the controls of pCO2 variations. The authors correctly write "In addition to influence of temperature, the spatiotemporal distribution of pCO2 in surface seawater is affected by the biological utilization of CO2, the vertical and lateral transport, the sea-air exchange of CO2 and terrestrial inputs." However, they do not quantify the relative importance of these controls in their data although there are published methods for such quantification (e.g. Olsen et al 2008). Specifically, the importance of fresh water input and air-sea exchange need to be quantified. This should be feasible since they have seasonal data of two parameters of the CO2-system in addition to SST, SSS, and nutrients. 2- The readability of the manuscript need to be improved. For instance, the study area is quite small, but quite complicated in terms of processes and interactions. Hence, there are a lot of names used in the manuscript (e.g. Gulf of Cadiz Current; AZORES Current; Guadalquivir River; Bay of Cádiz; Cape San Vicente), but locations of these are not shown anywhere in the manuscript. Including these names in the maps/figures would enhance the readability of the manuscript. It is also my opinion that it would be much easier to read the paper if the authors present results in seasonal maps (they do that for CO2 flux in Fig. 10) and then discuss the controls of pCO2 changes between seasons and places.

Specific comments: Line 19, "On the other side" do you mean "on the other hand?" Line 48, after "all other organisms" please add "which increases the concentration of inorganic carbon" Line 50 "generate uncertainty" please replace with "is not clearly defined" Lines 62-65 I do not understand the sentences between "Finally, the inner.." and "….towards offshore (Walsh 1991)." Line 193, "T values were significantly different among all cruises (p < 0.05)" why is this important result to mention? Line 96-97 "Spatially T tended to increase from coastal to offshore areas" during all seasons? or

during winter? Lines 211 – 215. I do not understand. Do you mean that both underway and discrete data are shown in Fig 2B? if so please clarify this in the caption and explain more about the reasons for differences between different data. Line 238 "TF presented the highest mean concentration for the whole study ($0.77 \pm 0.76$ $\mu$mol L-1)." I notice that given the mean PO4 of 0.28 this mean NO3 is much less than what is expected from Redfield, is this typical for the area? Lines 283 – 291, please state the uncertainty of the implied pCO2 growth. Please elaborate why you believe the excess pCO2 growth (over the atmospheric growth) is caused by continental input. Lines 300 – 3005, can the reason for difference pCO2 over different depth ranges be due to different TA/DIC ratios in the FW influenced areas and those offshore? Line 321, in which form is the CO2 input? Lines 333 - 334, How pCO2 increase can be computed from only F? or do you make more assumptions? Lines 335 – 342, you mention that upwelling systems can be influencing the distribution of pCO2 in the Gulf of Cadiz. BUT do you have any evidence for such influence in your data? If not why do you mention it here?

Figures: Figure 1: show important currents and places mentioned in the text. Figures 2, 3, 5, 6, and 7. Clarify in the caption whether both underway and discrete data are used.

Reference: Olsen, A. et al. (2008), Sea-surface CO2 fugacity in the subpolar North Atlantic, Biogeosciences, 5, 535-547, doi:10.5194/bg-5-535-2008.

---

## Author Comment (AC1) · 7 Jun 2019

Reviewer 1:

The authors are very grateful for your constructive comments and suggestions on the previous version. We think that the manuscript has been significantly improved thanks to all the contributions made. Below you will find the comments you made and our comments as authors (marked AC) on each point. In response to all the comments the manuscript has been modified, resulting in changes to line numbers. Therefore, we have included the new line numbers (whenever applicable) so that you can refer to either the current or (former) version if you wish.

Thank you very much for your consideration. Sincerely, Dolores Jiménez-López, on

behalf of all co-authors.

Jiménez-López et al. discuss spatio-temporal variability of pCO2 in the Gulf of Cádiz based on a new dataset collected between March 2014 and February 2016. These results will eventually help to better understand carbon cycle processes on continental shelves and their contribution to the global carbon cycle. And although the authors discuss many different and interesting local features, the submitted manuscript lacks clarity and should be revised and restructured before publication.

*General comments

The separation of the driving mechanisms of pCO2 into temperature and biological effects follows the line of argumentation of Takahashi et al. (2002), however, the authors actually calculate thermal and non-thermal components of pCO2 (e.g. Landschützer et al. (2015)). This is also stated by the authors themselves (Line 161 or 372), but not implemented or followed in their discussion, which is a consequence of the fact that the wording in the method section 2.3 is almost identical to the description of the method in Takahashi et al. (2002) (but not cited as such). Especially in the continental shelves, complex interactions of air-sea gas exchange, primary production, lateral and vertical transport, entrainment of high-DIC waters from below, anthropogenic runoff and freshwater addition lead to changes in salinity, DIC and alkalinity and thereby affect the non-thermal trend in pCO2. Moreover, the authors show seasonality, which they attribute to temperature and biological effects only, while at the same time, they discuss how, e.g., river runoff changes in magnitude over the year and thereby affects pCO2. Although the authors present many different drivers of pCO2 variability, they go back in the temperature-biology framework, which is inconsistent and difficult to follow.

The discussion section needs to be restructured accordingly. First, only results of the 8 cruises should be interpreted without repeating the results. Second, the results should be put in context with previous studies that took place in the same study area and its vicinity; here, it is crucial to include the reference years and seasons. And last, the

findings for the Gulf of Cádiz may be compared to other continental shelf areas in the North Atlantic and globally. At the moment, the authors list many results, in the result and discussion section, with little interpretation and no clear line of argumentation that leads to the presented conclusion.

AC: Thank you very much for your suggestions. We have changed the terms "temperature and biological effects" to "thermal and non-thermal" components of pCO2. In addition, we have determined the contribution of SST, air-sea CO2 exchange and mixing plus biological processes to the changes of pCO2 using the Olsen et al. (2008) method (suggested by the reviewer 3). This quantification appears in the Material and methods (Line 174-191) and Discussion sections (Line 374-391) and a new figure has been added (Fig. 7). The Discussion section has been modified following your suggestions. Firstly, some Discussion paragraphs are moved to the Results section (Line 223-229 and Line 244-249). Second, in the Discussion, the results are considered in the context of previous studies of the same area (including references to years and seasons) and then in the context of studies of other continental shelf areas. And last, Table 4 has been removed, and Table 5 is now Table 4, in which only studies of the Gulf of Cádiz are included.

*Specific comments

-Line 74: If previous studies have already determined the sink strength of the Gulf of Cádiz, seasonally driven by temperature and biology, what is the added value of your study? I am missing a clear motivation for this manuscript in the introduction.

AC: Thank you very much for your suggestion. We have included the following in the text to explain the added value that our study provides: Line 73-77: "It has also been possible to estimate the influence that various sea surface currents have on pCO2 variability, since this study considers deeper areas than previous works. Therefore, we can analyse the change that has occurred in relation to the CO2 uptake capacity in the Gulf of Cádiz in the last 10 years, in comparison with other studies that analyse the

seasonal variation underway of pCO2 in this area (Ribas-Ribas et al., 2011)".

-Line 84 / Figure 1: Add bathymetry. Add position of river Guadalete and the tidal creeks and the position of cape San Vicente. You may want to add a circulation scheme here that would help to visualise the surface circulation here.

AC: We followed your suggestions and edited Fig. 1. However the tidal creek named "River San Pedro" could not be added due to its small dimension compared to the rivers Guadalquivir and Guadalete.

-Line 110: Do the transects cover different water masses or circulation features? AC: Yes, the shallower stations of the different transects are influenced by the Gulf of Cádiz Current and the deeper stations (about 300 m approximately) by the Azores Current. In Fig. 1 the circulation scheme of the study area is illustrated. Line 133: How do you correct the temperature difference?

AC: Corrections between the equilibrator and SST were made following the method of Takahashi et al. (1993). The sentence has been edited in the text. Line 127-129: "The xCO2 was converted into pCO2 according to the protocol described in DOE (2007). Corrections between the equilibrator and SST were made following Takahashi et al. (1993). The temperature difference between the ship's sea inlet and the equilibrator was less than 1.5 °C".

-Line 144: How was the oxygen sensor calibrated? Confusing to first explain how AOU is derived, without a detailed description of how oxygen values were determined.

AC: This point is now clarified and added in the text. Line 134-138: "Dissolved oxygen values were obtained with the sensor of the rosette (SeaBird 63) pre-calibrated using Winkler titration ($\pm$0.1 $\mu$mol L-1) of samples collected from several water depths at selected stations (Parsons et al., 1984). Apparent Oxygen Utilization (AOU) was determined as the difference between the solubility calculated applying the expression proposed by Weiss (1974) and the experimental values of dissolved oxygen".

-Line 150: Why unit "mile"? Why is exactly 0.5 mile chosen? What is the distance between two stations? Particularly in the SP section, could there be an overlap in pCO2 data for calculating the mean? If there is a CTD coupled to the rosette-sampler, why are not discrete SST and SSS data used for each station and compared / evaluated to the underway SST / SSS measurements?

AC: We have used the unit "mile" to facilitate our estimation of transit times between the sampling stations. The mean distance between stations was set at 5 miles from the beginning, and then 0.5 mile constitutes approximately 10% of this distance. The mean pCO2 of the study was not calculated using the discrete values, but by the underway measurements. In any case, we have reviewed the manuscript and the data and we have observed that the SP7 station represented in the previous Fig. 1 is not included in the results presented in the Table 2. This station was removed in the new Fig. 1.

SST and SSS data for each station and for the underway measurements were compared and they do not show differences greater than 0.04 °C and 0.01 units, respectively. This point was clarified and added in the text: Line 146-148: "SST and SSS data were compared with the values collected with the CTD coupled to the rosette-sampler and they do not show differences greater than 0.04 °C and 0.01 units, respectively". Moreover, the discrete values obtained through the underway measurements of SST and SSS were used because they are necessary in subsequent calculations.

-Line 200: How can there be no spatial and seasonal variation in SSS, when there are freshwater inputs through storms and rivers?

AC: Spatial variation was only observed in the area of the Guadalquivir River related with a storm period that led to very heavy freshwater discharges during December 2014 (Line 230-231).

-Line 212: In which zones exactly? If sharp pCO2 variations are observed that coincide with discrete sampling stations, could that be related to the sampling strategy (e.g. potential sampling of ship exhaust) and not be a real signal? Do you correct for this?

[Figure]

How can the sampling time be depth-dependent if only discrete samples were taken at 5m depth (Line 135)?

AC: The sampling time in each station was variable because water samples were taken at different depths of the water column with Niskin bottles, which were mounted on a rosette-sampler, although in this study we use only the "surface" sample at 5 m. In addition, as these cruises were multidisciplinary, the sampling time was dependent on the various activities carried out in each station. For example, at some stations, this activity could take up to 8 hours due to the sampling of zooplankton where a bongo and neuston net and/or multinet was used. Studies such as those of Sierra et al. (2017a, 2017b), González-García et al. (2018) are examples of these cruises and the other activities carried out here. This sentence is now clarified in the text: Line 239-241: "In Fig. 2 a sharp variation of SST and pCO2 can be observed in some zones that coincides with the stations where discrete water samples were taken. This may be due to the different sampling time at these stations, which varied between 2 and 8 hours in function of the depth of the system". Additionally, we have observed some instability of the underway measurements of pCO2 in the areas that coincide with the position of the discrete water samples (Fig. 1), due to changes in the flow pump of the ship when its dynamic positioning was functioning.

-Line 220-228: Are there no spatial differences in pH and AOU?

AC: There are some spatial differences in pH and AOU, although a general trend was not observed. They seem related more to the intensity of local processes, such as continental inputs through the River Guadalquivir, increase of the primary production in coastal areas, influence of the upwelling in Trafalgar and relative change in the intensity of the surface currents. A short sentence is now added in the text: Line 259: "No general trend in the spatial variations of pH and AOU was found".

-Line 258: Is it truly equivalent? 17.4 $\mu$atm C-1 divided by 400 $\mu$atm results in 0.0435C-1.

AC: Our apologies, the correct value is 16.9 $\mu$atm °C-1. This has been rectified in the manuscript (Line 289). Thank you for drawing this to our attention.

-Line 265: It is not clear to me, why table 4 is useful. Clearly, local effects and seasonality impact pCO2-SST relationships, but they are not discussed or put in perspective with the results of the Gulf of Cádiz.

AC: Yes, you are right. We have removed Table 4, and we have discussed in the text certain relationships and seasonal variations found in other studies.

-Line 290: The larger trend in pCO2 in the ocean than in the atmosphere can be driven by

AC: We are sorry, but we think that this suggestion is not complete. In any case, this sentence has been clarified in the manuscript: Line 311-314: "This suggests a possible increase of the anthropogenic nutrient and C inputs from land (Mackenzie et al., 2004) since the direction and magnitude of estuarine and continental shelf CO2 exchange with the atmosphere is highly dependent on the terrestrial organic budget and nutrient supplies to the coastal ocean (Borges and Abril, 2011; Cai, 2011)".

-Line 300-305: There is no statistical difference in pCO2 or temperature with bottom depth, which might be because Figure 5 shows data from all seasons and years.

AC: Yes, you are correct. With this figure, we want to show only the general trend of pCO2 and SST at different intervals of depth of the water column through offshore areas. Fig. 5 has been modified to Fig. 4, and this paragraph moved to Results, now Line 242-247.

-Line 362: You only show the relationship between AOU or pH and pCO2 but there is no discussion of it. Why is almost the entire study area over different seasons oversaturated in oxygen?

AC: This point is now explained and discussed better in the following paragraph of the text (Line 361-373). The oversaturation in oxygen may be due to the influence

of two factors: the first is, greater photosynthetic activity in the area throughout the year (González-García et al., 2018) and the air injection processes responsible for an mean increase of $7\mu$mol L-1 in the surface waters of the ocean (Sarmiento and Gruber, 2006).

-Line 377: total or mean T/B. The T/B ratios by Ribas-Ribas et al. (2011) and de la Paz et al. (2009) have been estimated for which years or seasons?

AC: It is total T/B. Years and seasons of these studies are included in Table 4, and in the text we have included a notification to refer to this table.

-Line 382: How does the DIC flux from the sediment affect T/B?

AC: Benthic DIC flux is another source of inorganic carbon that increases the $CO_2$ concentration in the water column, which would affect the increase of $CO_2$ non-thermal.

-Line 385: What is the cause for $\Delta$pCO2bio variations over depth?

AC: The variations of $\Delta$pCO2bio (now modified to $\Delta$pCO2 non-thermal) observed with respect to the system depth are due to the influence of several processes. In areas close to the coast there is an increase of $\Delta$pCO2non-thermal due to continental inputs, greater primary production and the remineralization of the organic matter in the sediment. In the central area, there is a decrease of these 3 processes. And in the deepest areas, there is an increase of $\Delta$pCO2 non-thermal with the change in the origin of the surface currents. This point clarified in the text, Line 404-419.

-Line 389: If $\Delta$pCO2temp and bio are calculated as a seasonal amplitude, what temperature and chlorophyll values are used to establish the dependency here? Are these annual means (same for Figure 8 A and B). In any case, I do not understand how the thermal component in relation to temperature and the non-thermal component in relation to chlorophyll confirms the importance of different processes on pCO2 variation.

AC: Temperature and chlorophyll values used to establish the dependency are the mean values of the 8 cruises for each of the discrete sampling stations. Following

the suggestions of other reviewers, Fig. 8 was removed and the importance of different processes on pCO2 variation was calculated by a different method (Olsen et al., 2008), (Line 374-391 and Fig. 7).

-Line 397: How can the surface chlorophyll and nutrients be constant, when there is a large gradient with distance to the coast (Line 395)?

AC: Thank you for this question. There was a mistake in the text and it has been corrected (Line 410). We wanted say that chlorophyll-a and nutrients concentrations decrease exponentially with the depth system, but their values are relatively constant in waters with bottom-depth higher to 200 m.

-Line 405: Are the T/B ratios for the different transects significantly different from each other?

AC: T/B ratios for the different transects are not associated with a standard deviation since they are calculated as total ratios, so is not possible to determine significant differences between transects.

-Line 422: Again, why is table 5 helpful? I understand that there are many studies that evaluate shelf area processes in the North Atlantic, but this is not discussed in the manuscript. It appears more as a list of literature than it helps to put you own results in perspective.

AC: Yes, you are right. Table 5 is now Table 4 and following your suggestion it has been modified. In this table only the studies carried out in the Gulf of Cádiz are included, and these are also discussed in the manuscript. Other general studies are also discussed in the text where relevant.

-Figure 3: Panels should have the same size; panel B should next to or below panel A. Add linear correlation equation including units for both panels.

AC: Corrected.

-Figure 6: Why are there 2 regression lines plotted for AOA-pCO2?

AC: Corrected.

-Figure 7 / 9: What are the uncertainties of the thermal and non-thermal components? Are they significantly different from each other?

AC: T/B is a total ratio, so it is not associated with a standard deviation; nor is it possible to determine significant differences between components either. Figure 9 is now Figure 8.

-Figure 10: You could change the colourbar; it is not clear where the border between outgassing and uptake is located.

AC: The border between outgassing and uptake is shown in yellow now (Fig. 10).

-You could simply state in section 2.5. that all reported linear correlations are statistical significant with p-values smaller than 0.05 in the entire manuscript unless indicated otherwise. With that, you do not have to report the p-value again. There a numerous linear relationship equations in the manuscript without units. The correlation equations could be plotted within the according figures to increase readability.

AC: Corrected, thank you for your suggestions. Section 2.5 modified, Line 207-209: "The threshold value for statistical significance was taken as $p < 0.05$. Moreover, all reported linear correlations are type I and they are statistically significant with p-values smaller than 0.05 in the entire manuscript unless indicated otherwise". Units in the linear relationship are included in the text. Correlation equations have been plotted within the figures but without units, since there is insufficient space (Fig. 5, 6 and 11).

-Please have colour blind people in mind for all figures. It is not possible to differentiate between years with the presently used lighter / darker colours; you could use different symbols as well.

AC: Fig. 5, 6 and 11 have were modified using different symbols.

-Abbreviations that are only used in one paragraph only are superfluous; for example EBUS. Consider abbreviating T by SST and and S by SSS for readability.

AC: These suggestions have been considered in the text.

-The manuscript will benefit from the input of a native speaker. There is a need to check for incomplete sentences and the use of correct tenses. There should be fewer, but longer paragraphs that consist of more than one or two sentences; while covering the same topic. This will make it easier to follow clear arguments.

AC: A native speaker with experience of scientific papers has revised the manuscript again.

References:

González-García, C., Forja, J., González-Cabrera, M. C., Jiménez, M. P., and Lubián, L. M.: Annual variations of total and fractionated chlorophyll and phytoplankton groups in the Gulf of Cádiz, Sci. Total Environ., 613, 1551–1565, https://doi.org/10.1016/j.scitotenv.2017.08.292, 2018.

Olsen, A., Brown, K. R., Chierici, M., Johannessen, T., Neill, C.: Sea-surface CO2 fugacity in the subpolar North Atlantic, Biogeosciences, 5, 535-547, https://doi.org/10.5194/bg-5-535-2008, 2008.

Sierra, A., Jiménez-López, D., Ortega, T., Ponce, R., Bellanco, M. J., Sánchez-Leal, R., Gómez-Parra, A., and Forja, J.: Spatial and seasonal variability of CH4 in the eastern Gulf of Cadiz (SW Iberian Peninsula), Science of the Total Environment, 590, 695-707, 2017a.

Sierra, A., Jiménez-López, D., Ortega, T., Ponce, R., Bellanco, M. J., Sánchez-Leal, R., Gómez-Parra, A., and Forja, J.: Distribution of N2O in the eastern shelf of the Gulf of Cadiz (SW Iberian Peninsula), Science of the Total Environment, 593, 796-808, http://dx.doi.org/10.1016/j.scitotenv.2017.03.189, 2017b.

Sarmiento, J. L., and Gruber, N.: Ocean biogeochemical dynamics. Princeton Univ. Press., 2006.

Please also note the supplement to this comment:
https://www.ocean-sci-discuss.net/os-2019-6/os-2019-6-AC1-supplement.zip

---

## Author Comment (AC2) · 7 Jun 2019

Reviewer 2:

The authors are very grateful for your constructive comments and suggestions on the previous version. We think that the manuscript has been significantly improved thanks to all the contributions made. Below you will find the comments you made and our comments as authors (marked AC) on each point. In response to all the comments the manuscript has been modified, resulting in changes to line numbers. Therefore, we have included the new line numbers (whenever applicable) so that you can refer to either the current or (former) version if you wish.

Thank you very much for your consideration. Sincerely, Dolores Jiménez-López, on

behalf of all co-authors.

-In general, the manuscript is written with too many very short paragraphs.

AC: An experienced native speaker revised the manuscript again to ensure that paragraph lengths are appropriated to the content and style, and generally to improve the manuscript.

-The topic of the presented research is interesting and pertinent in the actual context of climate change. The title is very appealing and the reader expects to learn about the processes that control the $pCO_2$ variability in the eastern Gulf of Cadiz. Unfortunately, the abstract does not reveal any factor controlling the $pCO_2$ variability. The conclusion indicates that: "temperature and biological activity are the two principal factors that explain the temporal variability of CO2". This is the case everywhere in the ocean. There is nothing new here! Then (line 470), it is indicated that: ": : : the distribution is principally controlled by the temperature". Here again, there is nothing new and this is normal. Therefore, I would suggest the authors to change the title of their manuscript to more accurately reflect its content.

AC: Thank for your suggestion. The title has been modified to "pCO2 variability in surface waters of the eastern Gulf of Cádiz (SW Iberian Peninsula)".

-The authors should also precise (and justify) what kind of linear regression (type I, type II, : : :) they did in order to determine the general trends.

AC: We have added the kind of linear regression in the text (type I, Line 208).

-They should also detail all their calculations (including simple ones such as the mean capture capacity, : : :).

AC: The mean capture capacity is calculated using the total surface of the study area (52.8•102 km2) and the mean annual flux during the 8 cruises (-0.18 mmol m-2 d-1). This detail is now added in the manuscript (Line 466-467). Another calculation not indicated in the text and that may be of interest is the mean benthic flux of CO2 (Line

342), which is calculated making certain assumptions that are stated in the text (Line 343-346).

-The last sentence should be re-written to me more precise (what means ": : :variability of CO2: : :"; is it TCO2, CO2 flux, pCO2???; what means ": : :capacity of CO2 capture: : :"; surface/depth, on what timescale?). The manuscript will gain in clarity if it were much concise.

AC: This final paragraph has been modified to give more clarity. Line 491-494: "The annual uptake capacity of CO2 by the surface waters in our study area is 14.9 Gg C year-1. The CO2 fluxes present seasonal variation: these waters act as a source of CO2 to the atmosphere in summer and autumn and as a sink in winter and spring. Based on the information available in the zone, there seems to have been a decrease in the capacity for CO2 capture in the zone in recent decades".

-There is a need to remove some of the tables and figures.

AC: Thank you for your suggestion. Fig. 8 and Table 4 have been removed. Table 5 is now Table 4 and only includes studies in the Gulf of Cádiz. Fig. 1 has been improved.

-In summary, this manuscript needs a major revision before publishing.

AC: Agreed. We hope the substantial revisions we have made to the manuscript now make the paper suitable for publication.

References:

Olsen, A., Brown, K. R., Chierici, M., Johannessen, T., Neill, C.: Sea-surface CO2 fugacity in the subpolar North Atlantic, Biogeosciences, 5, 535-547, https://doi.org/10.5194/bg-5-535-2008, 2008.

Please also note the supplement to this comment:
https://www.ocean-sci-discuss.net/os-2019-6/os-2019-6-AC2-supplement.zip

---

## Author Comment (AC3) · 7 Jun 2019

Reviewer 3:

The authors are very grateful for your constructive comments and suggestions on the previous version. We think that the manuscript has been significantly improved thanks to all the contributions made. Below you will find the comments you made and our comments as authors (marked AC) on each point. In response to all the comments the manuscript has been modified, resulting in changes to line numbers. Therefore, we have included the new line numbers (whenever applicable) so that you can refer to either the current or (former) version if you wish.

Thank you very much for your consideration. Sincerely, Dolores Jiménez-López, on

behalf of all co-authors.

The authors investigate factors controlling pCO2 variations in the Gulf of Cadiz. They use high quality data from 8 cruises incorporating underway data of pCO2, SSS, SST, and wind speed as well as discrete data for pH, AOU, and nutrients taken along three repeat transects during the cruises. They present spatiotemporal distributions of the underway data, the cruise averages of the discrete data, and the seasonal changes of the computed air-sea CO2 flux. The authors then discuss the factors influencing the pCO2 variability. Specifically, they quantify thermal/non-thermal controls of pCO2. They conclude that temperature and biological activity are the two principal factors that explain the temporal variability of pCO2. They also point out that continental inputs and mixing with water originating from warm ocean currents influence the spatial variability of pCO2.

The work is OK structured, includes original research based on high quality data, and suits for publication in this journal. However, there are several things that need improvements and/or clarification and I recommend major revision.

*General comments

-1- The main subject of the study is the controls of pCO2 variations. The authors correctly write "In addition to influence of temperature, the spatiotemporal distribution of pCO2 in surface seawater is affected by the biological utilization of CO2, the vertical and lateral transport, the sea-air exchange of CO2 and terrestrial inputs." However, they do not quantify the relative importance of these controls in their data although there are published methods for such quantification (e.g. Olsen et al 2008). Specifically, the importance of fresh water input and air-sea exchange need to be quantified. This should be feasible since they have seasonal data of two parameters of the CO2-system in addition to SST, SSS, and nutrients.

AC: Thank you very much for your suggestion, which is very interesting. The Olsen et al. (2008) method has been taken into account in the revised manuscript and the

contributions of SST, air-sea CO2 exchange and mixing plus biology on pCO2 change have been quantified. This method also considers change due to SSS variations but we have not included this quantification, since we do not have available the variations of total alkalinity and dissolved inorganic carbon, and the spatial changes of SSS are only significant near the Guadalquivir River mouth (a point included in the text, Line 186-187). The manuscript has been edited in Material and methods (Line 174-191) and Discussion sections (Line 374-391) to include this quantification. A new figure has been added (Figure 7).

-2- The readability of the manuscript need to be improved. For instance, the study area is quite small, but quite complicated in terms of processes and interactions. Hence, there are a lot of names used in the manuscript (e.g. Gulf of Cadiz Current; AZORES Current; Guadalquivir River; Bay of Cádiz; Cape San Vicente), but locations of these are not shown anywhere in the manuscript. Including these names in the maps/figures would enhance the readability of the manuscript. It is also my opinion that it would be much easier to read the paper if the authors present results in seasonal maps (they do that for CO2 flux in Fig. 10) and then discuss the controls of pCO2 changes between seasons and places.

AC: Thank you for your suggestion. Fig. 1 has been edited to add different processes and names used to improve the readability of the manuscript. However, it is not possible to present results in seasonal maps,except for the spatial distribution of CO2 fluxes, since for example in Fig. 2, the data interpolated in our database for the same season would not be right.

*Specific comments

-Line 19, "On the other side" do you mean "on the other hand?"

AC: Corrected.

-Line 48, after "all other organisms" please add "which increases the concentration of

inorganic carbon"

AC: This point has been added in the text, thank you (Line 46).

-Line 50 "generate uncertainty" please replace with "is not clearly defined"

AC: Corrected.

-Lines 62-65 I do not understand the sentences between "Finally, the inner.." and ": : :.towards offshore (Walsh 1991)."

AC: These sentences have been modified (Line 57-62). The effect of the continental inputs on pCO2 variation are explained.

-Line 193, "T values were significantly different among all cruises (p < 0.05)" why is this important result to mention?

AC: Yes, you are correct. In any case, p-values have been removed from the manuscript and a brief reference has been added to the Statistical analysis section. Line 207-209: "The threshold value for statistical significance was taken as p < 0.05. Moreover, all reported linear correlations are type I and they are statistically significant with p-values smaller than 0.05 in the entire manuscript unless indicated otherwise".

-Line 96-97 "Spatially T tended to increase from coastal to offshore areas" during all seasons? Or during winter?

AC: This sentence has been modified in the text. It was not clear in the previous version of the manuscript. Line 217-218: "In general, spatially SST tended to increase from coastal to offshore areas during spring and winter, while in summer and autumn this SST gradient was inverse (Fig. 2A)".

-Lines 211-215. I do not understand. Do you mean that both underway and discrete data are shown in Fig 2B? if so please clarify this in the caption and explain more about the reasons for differences between different data.

AC: Fig. 2B only show the underway measurements, but we have observed an increase of these pCO2 values in the areas that coincide with the position of the discrete water samples (Fig. 1). This sentence has been edited in the manuscript: Line 239-241: "In Fig. 2 a sharp variation of SST and pCO2 can be observed in some zones that coincides with the stations where discrete water samples were taken. This may be due to the different sampling time at these stations, which varied between 2 and 8 hours in function of the depth of the system".

-Line 238 "TF presented the highest mean concentration for the whole study (0.77 _ 0.76 _mol L-1)." I notice that given the mean PO4 of 0.28 this mean NO3 is much less than what is expected from Redfield, is this typical for the area?

AC: Low N/P relationships are typical for this study area (Anfuso et al., 2010). This information has been added in the text: Line 273-275: "The mean N/P ratio in surface waters for the whole study was 3.5 ± 2.0, similar to that estimated by Anfuso et al. (2010) in the northeast continental shelf of the Gulf of Cádiz, which indicates a relative phosphate deficit with respect to the Redfield ratio (Redfield et al., 1963)".

-Lines 283-291, please state the uncertainty of the implied pCO2 growth. Please elaborate why you believe the excess pCO2 growth (over the atmospheric growth) is caused by continental input.

AC: The uncertainty value has been added in the text (Line 309). The pCO2 growth caused by continental inputs was also found by other authors and this point is included in the text. Line 311-314: "This suggests a possible increase of the anthropogenic nutrient and C inputs from land (Mackenzie et al., 2004) since the direction and magnitude of estuarine and continental shelf CO2 exchange with the atmosphere is highly dependent on the terrestrial organic budget and nutrient supplies to the coastal ocean (Borges and Abril, 2011; Cai, 2011)".

-Lines 300-305, can the reason for difference pCO2 over different depth ranges be due to different TA/DIC ratios in the FW influenced areas and those offshore?

AC: Thank you for your comment. This is a very interesting question. However, at the moment, we do not have available this information needed to answer it.

-Line 321, in which form is the CO2 input?

AC: The form of CO2 input is inorganic carbon. This sentence has been edited in the text (Line 331-334).

-Lines 333-334, How pCO2 increase can be computed from only F? or do you make more assumptions?

AC: More assumptions are necessary: mean depth of the water column (23 m) and the fact that it is well-mixed; a pH = 8; in the conditions of mean temperature and salinity in the Gulf of Cádiz (18.8 °C and 36.19, respectively) and using the K1 and K2 acidity constants proposed by Lueker et al. (2000) in the total pH scale (information indicated in the text: Line 341-346).

-Lines 335-342, you mention that upwelling systems can be influencing the distribution of pCO2 in the Gulf of Cadiz. BUT do you have any evidence for such influence in your data? If not why do you mention it here?

AC: There is some evidence in our data for the Trafalgar transect. This point is included in the text: Line 351- 354: "In our database experimental evidence of the upwelling was found only in the TF transect. A local decrease of the mean values of SST (17.4 °C) and pCO2 (399.1 $\mu$atm) was observed in this coastal area of TF, with respect to the deeper areas (18.8 °C and 405.1 $\mu$atm, respectively) for the whole period". In addition, in Fig. 10 it can also observed that the areas near to the Trafalgar section show lower values of CO2 flux during summer and winter.

-Figures: Figure 1: show important currents and places mentioned in the text. Figures 2, 3, 5, 6, and 7. Clarify in the caption whether both underway and discrete data are used.

AC: Fig. 1 has been improved and the figure captions have been clarified.

References:

Anfuso, E., Ponce, R., Castro, C. G., and Forja, J. M.: Coupling between the thermo-haline, chemical and biological fields during summer 2006 in the northeast continental shelf of the Gulf of Cádiz (SW Iberian Peninsula), 47–56, Sci. Mar., https://doi: 10.3989/scimar.2010.74s1047, 2010.

Please also note the supplement to this comment:
https://www.ocean-sci-discuss.net/os-2019-6/os-2019-6-AC3-supplement.zip

---

## Author Response (AR1)

Dear Editor:

The authors are very grateful to the three anonymous reviewers for their constructive comments and suggestions in the previous version. We have addressed the reviewers' annotations and revised carefully the manuscript in order to improve our work. In this revised version, several mistakes pointed out by the reviewers have been corrected. We think that the manuscript has been significantly improved thanks to all the contributions made.

Below you will find the comments they made and our comments as authors (marked AC) on each point. In response to all the comments the manuscript has been modified, resulting in changes to line numbers. Therefore, we have included the new line numbers (whenever applicable) so that you can refer to either the current or (former) version if you wish.

Thank you very much for your consideration.

Sincerely,

Dolores Jiménez-López, on behalf of all co-authors.

* **Reviewer 1:**

*Jiménez-López et al. discuss spatio-temporal variability of pCO₂ in the Gulf of Cádiz based on a new dataset collected between March 2014 and February 2016. These results will eventually help to better understand carbon cycle processes on continental shelves and their contribution to the global carbon cycle. And although the authors discuss many different and interesting local features, the submitted manuscript lacks clarity and should be revised and restructured before publication.*

- *General comments*

*The separation of the driving mechanisms of pCO₂ into temperature and biological effects follows the line of argumentation of Takahashi et al. (2002), however, the authors actually calculate thermal and non-thermal components of pCO₂ (e.g. Landschützer et al. (2015)). This is also stated by the authors themselves (Line 161 or 372), but not implemented or followed in their discussion, which is a consequence of the fact that the wording in the method section 2.3 is almost identical to the description of the method in Takahashi et al. (2002) (but not cited as such). Especially in the continental shelves, complex interactions of air-sea gas exchange, primary production, lateral and vertical transport, entrainment of high-DIC waters from below, anthropogenic runoff and freshwater addition lead to changes in salinity, DIC and alkalinity and thereby affect the non-thermal trend in pCO₂. Moreover, the authors show seasonality, which they attribute to temperature and biological effects only, while at the same time, they discuss how, e.g., river runoff changes in magnitude over the year and thereby affects pCO₂. Although the authors present many different drivers of pCO₂ variability, they go back in the temperature-biology framework, which is inconsistent and difficult to follow.*

*The discussion section needs to be restructured accordingly. First, only results of the 8 cruises should be interpreted without repeating the results. Second, the results should be put in context with previous studies that took place in the same study area and its vicinity; here, it is crucial to include the reference years and seasons. And last, the findings for the Gulf of Cádiz may be compared to other continental shelf areas in the North Atlantic and globally. At the moment, the authors list many results, in the result and discussion section, with little interpretation and no clear line of argumentation that leads to the presented conclusion.*

AC: Thank you very much for your suggestions. We have changed the terms "temperature and biological effects" to "thermal and non-thermal" components of pCO₂. In addition, we have determined the contribution of SST, air-sea CO₂ exchange and mixing plus biological processes to the changes of pCO₂ using the Olsen et al. (2008) method (suggested by the reviewer 3). This quantification appears in the Material and methods (Line 174-191) and Discussion sections (Line 374-391) and a new figure has been added (Fig. 7).

The Discussion section has been modified following your suggestions. Firstly, some Discussion paragraphs are moved to the Results section (Line 223-229 and Line 244-249). Second, in the Discussion, the results are considered in the context of previous studies of the same area (including references to years and seasons) and then in the context of studies of other continental shelf areas. And last, Table 4 has been removed, and Table 5 is now Table 4, in which only studies of the Gulf of Cádiz are included.

          -   *Specific comments*

*Line 74: If previous studies have already determined the sink strength of the Gulf of Cádiz, seasonally driven by temperature and biology, what is the added value of your study? I am missing a clear motivation for this manuscript in the introduction.*

75

AC: Thank you very much for your suggestion. We have included the following in the text to explain the added value that our study provides:

Line 73-77: "It has also been possible to estimate the influence that various sea surface currents have on $pCO_2$ variability, since this study considers deeper areas than previous works. Therefore, we can analyse the change that has occurred in relation to the $CO_2$ uptake capacity in the Gulf of Cádiz in the last 10 years, in comparison with other studies that analyse the seasonal variation underway of $pCO_2$ in this area (Ribas-Ribas et al., 2011)".

*Line 84 / Figure 1: Add bathymetry. Add position of river Guadalete and the tidal creeks and the position of cape San Vicente. You may want to add a circulation scheme here that would help to visualise the surface circulation here.*

AC: We followed your suggestions and edited Fig. 1. However the tidal creek named "River San Pedro" could not be added due to its small dimension compared to the rivers Guadalquivir and Guadalete.

90

*Line 110: Do the transects cover different water masses or circulation features*?

AC: Yes, the shallower stations of the different transects are influenced by the Gulf of Cádiz Current and the deeper stations (about 300 m approximately) by the Azores Current. In Fig. 1 the circulation scheme of the study area is illustrated.

95   *Line 133: How do you correct the temperature difference?*

AC: Corrections between the equilibrator and SST were made following the method of Takahashi et al. (1993). The sentence has been edited in the text.

Line 127-129: "The $xCO_2$ was converted into $pCO_2$ according to the protocol described in DOE (2007). Corrections between the equilibrator and SST were made following Takahashi et al. (1993). The temperature difference between the ship's sea inlet and the equilibrator was less than 1.5 ºC".

*Line 144: How was the oxygen sensor calibrated? Confusing to first explain how AOU is derived, without a detailed description of how oxygen values were determined.*

105

AC: This point is now clarified and added in the text.

Line 134-138: "Dissolved oxygen values were obtained with the sensor of the rosette (SeaBird 63) pre-calibrated using Winkler titration ($\pm 0.1$ µmol $L^{-1}$) of samples collected from several water depths at selected stations (Parsons et al., 1984). Apparent Oxygen Utilization (AOU) was determined as the difference between the solubility calculated applying the expression proposed by Weiss (1974) and the experimental values of dissolved oxygen".

115

*Line 150: Why unit "mile"? Why is exactly 0.5 mile chosen? What is the distance between two stations? Particularly in the SP section, could there be an overlap in pCO₂ data for calculating the mean? If there is a CTD coupled to the rosette-sampler, why are not discrete SST and SSS data used for each station and compared / evaluated to the underway SST / SSS measurements?*

AC: We have used the unit "mile" to facilitate our estimation of transit times between the sampling stations.

The mean distance between stations was set at 5 miles from the beginning, and then 0.5 mile constitutes approximately 10% of this distance.

The mean $pCO_2$ of the study was not calculated using the discrete values, but by the underway measurements. In any case, we have reviewed the manuscript and the data and we have observed that the SP7 station represented in the previous Fig. 1 is not included in the results presented in the Table 2. This station was removed in the new Fig. 1.

SST and SSS data for each station and for the underway measurements were compared and they do not show differences greater than 0.04 ℃ and 0.01 units, respectively. This point was clarified and added in the text: Line 146-148: "SST and SSS data were compared with the values collected with the CTD coupled to the rosette-sampler and they do not show differences greater than 0.04 ℃ and 0.01 units, respectively". Moreover, the discrete values obtained through the underway measurements of SST and SSS were used because they are necessary in subsequent calculations.

*Line 200: How can there be no spatial and seasonal variation in SSS, when there are freshwater inputs through storms and rivers?*

AC: Spatial variation was only observed in the area of the Guadalquivir River related with a storm period that led to very heavy freshwater discharges during December 2014 (Line 230-231).

*Line 212: In which zones exactly? If sharp pCO₂ variations are observed that coincide with discrete sampling stations, could that be related to the sampling strategy (e.g. potential sampling of ship exhaust) and not be a real signal? Do you correct for this? How can the sampling time be depth-dependent if only discrete samples were taken at 5m depth (Line 135)?*

AC: The sampling time in each station was variable because water samples were taken at different depths of the water column with Niskin bottles, which were mounted on a rosette-sampler, although in this study we use only the "surface" sample at 5 m. In addition, as these cruises were multidisciplinary, the sampling time was dependent on the various activities carried out in each station. For example, at some stations, this activity could take up to 8 hours due to the sampling of zooplankton where a bongo and neuston net and/or multinet was used. Studies such as those of Sierra et al. (2017a, 2017b), González-García et al. (2018) are examples of these cruises and the other activities carried out here.

This sentence is now clarified in the text: Line 239-241: "In Fig. 2 a sharp variation of SST and $pCO_2$ can be observed in some zones that coincides with the stations where discrete water samples were taken. This may be due to the different sampling time at these stations, which varied between 2 and 8 hours in function of the depth of the system".

Additionally, we have observed some instability of the underway measurements of $pCO_2$ in the areas that coincide with the position of the discrete water samples (Fig. 1), due to changes in the flow pump of the ship when its dynamic positioning was functioning.

*Line 220-228: Are there no spatial differences in pH and AOU?*

AC: There are some spatial differences in pH and AOU, although a general trend was not observed. They seem related more to the intensity of local processes, such as continental inputs through the River

Guadalquivir, increase of the primary production in coastal areas, influence of the upwelling in Trafalgar and relative change in the intensity of the surface currents.

A short sentence is now added in the text: Line 259: "No general trend in the spatial variations of pH and AOU was found".

*Line 258: Is it truly equivalent? 17.4 µatm $C^{-1}$ divided by 400 µatm results in $0.0435 C^{-1}$.*

AC: Our apologies, the correct value is 16.9 µatm $°C^{-1}$. This has been rectified in the manuscript (Line 289). Thank you for drawing this to our attention.

*Line 265: It is not clear to me, why table 4 is useful. Clearly, local effects and seasonality impact $pCO_2$-SST relationships, but they are not discussed or put in perspective with the results of the Gulf of Cádiz.*

AC: Yes, you are right. We have removed Table 4, and we have discussed in the text certain relationships and seasonal variations found in other studies.

*Line 290: The larger trend in $pCO_2$ in the ocean than in the atmosphere can be driven by*

AC: We are sorry, but we think that this suggestion is not complete. In any case, this sentence has been clarified in the manuscript: Line 311-314: "This suggests a possible increase of the anthropogenic nutrient and C inputs from land (Mackenzie et al., 2004) since the direction and magnitude of estuarine and continental shelf $CO_2$ exchange with the atmosphere is highly dependent on the terrestrial organic budget and nutrient supplies to the coastal ocean (Borges and Abril, 2011; Cai, 2011)".

*Line 300-305: There is no statistical difference in $pCO_2$ or temperature with bottom depth, which might be because Figure 5 shows data from all seasons and years.*

AC: Yes, you are correct. With this figure, we want to show only the general trend of $pCO_2$ and SST at different intervals of depth of the water column through offshore areas. Fig. 5 has been modified to Fig. 4, and this paragraph moved to Results, now Line 242-247.

*Line 362: You only show the relationship between AOU or pH and $pCO_2$ but there is no discussion of it. Why is almost the entire study area over different seasons oversaturated in oxygen?*

AC: This point is now explained and discussed better in the following paragraph of the text (Line 361-373). The oversaturation in oxygen may be due to the influence of two factors: the first is, greater photosynthetic activity in the area throughout the year (González-García et al., 2018) and the air injection processes responsible for an mean increase of 7µmol $L^{-1}$ in the surface waters of the ocean (Sarmiento and Gruber, 2006).

*Line 377: total or mean T/B. The T/B ratios by Ribas-Ribas et al. (2011) and de la Paz et al. (2009) have been estimated for which years or seasons?*

AC: It is total T/B. Years and seasons of these studies are included in Table 4, and in the text we have included a notification to refer to this table.

*Line 382: How does the DIC flux from the sediment affect T/B?*

AC: Benthic DIC flux is another source of inorganic carbon that increases the $CO_2$ concentration in the water column, which would affect the increase of $CO_2$ non-thermal.

*Line 385: What is the cause for $\Delta pCO_2bio$ variations over depth?*

AC: The variations of $\Delta pCO_2bio$ (now modified to $\Delta pCO_2$ non-thermal) observed with respect to the system depth are due to the influence of several processes. In areas close to the coast there is an increase of $\Delta pCO_2$non-thermal due to continental inputs, greater primary production and the remineralization of the organic matter in the sediment. In the central area, there is a decrease of these 3 processes. And in the deepest areas, there is an increase of $\Delta pCO_2$ non-thermal with the change in the origin of the surface currents.
This point clarified in the text, Line 404-419.

*Line 389: If $\Delta pCO_2temp$ and bio are calculated as a seasonal amplitude, what temperature and chlorophyll values are used to establish the dependency here? Are these annual means (same for Figure 8 A and B). In any case, I do not understand how the thermal component in relation to temperature and the non-thermal component in relation to chlorophyll confirms the importance of different processes on $pCO_2$ variation.*

AC: Temperature and chlorophyll values used to establish the dependency are the mean values of the 8 cruises for each of the discrete sampling stations.
Following the suggestions of other reviewers, Fig. 8 was removed and the importance of different processes on $pCO_2$ variation was calculated by a different method (Olsen et al., 2008), (Line 374-391 and Fig. 7).

*Line 397: How can the surface chlorophyll and nutrients be constant, when there is a large gradient with distance to the coast (Line 395)?*

AC: Thank you for this question. There was a mistake in the text and it has been corrected (Line 410). We wanted say that chlorophyll-a and nutrients concentrations decrease exponentially with the depth system, but their values are relatively constant in waters with bottom-depth higher to 200 m.

*Line 405: Are the T/B ratios for the different transects significantly different from each other?*

AC: T/B ratios for the different transects are not associated with a standard deviation since they are calculated as total ratios, so is not possible to determine significant differences between transects.

*Line 422: Again, why is table 5 helpful? I understand that there are many studies that evaluate shelf area processes in the North Atlantic, but this is not discussed in the manuscript. It appears more as a list of literature than it helps to put you own results in perspective.*

AC: Yes, you are right. Table 5 is now Table 4 and following your suggestion it has been modified. In this table only the studies carried out in the Gulf of Cádiz are included, and these are also discussed in the manuscript. Other general studies are also discussed in the text where relevant.

*Figure 3: Panels should have the same size; panel B should next to or below panel A. Add linear correlation equation including units for both panels.*

AC: Corrected.

*Figure 6: Why are there 2 regression lines plotted for AOA-$pCO_2$?*

AC: Corrected.

*Figure 7 / 9: What are the uncertainties of the thermal and non-thermal components? Are they significantly different from each other?*

AC: T/B is a total ratio, so it is not associated with a standard deviation; nor is it possible to determine significant differences between components either. Figure 9 is now Figure 8.

*Figure 10: You could change the colourbar; it is not clear where the border between outgassing and uptake is located.*

AC: The border between outgassing and uptake is shown in yellow now (Fig. 10).

*You could simply state in section 2.5. that all reported linear correlations are statistical significant with p-values smaller than 0.05 in the entire manuscript unless indicated otherwise. With that, you do not have to report the p-value again. There a numerous linear relationship equations in the manuscript without units. The correlation equations could be plotted within the according figures to increase readability.*

AC: Corrected, thank you for your suggestions.
Section 2.5 modified, Line 207-209: "The threshold value for statistical significance was taken as $p < 0.05$. Moreover, all reported linear correlations are type I and they are statistically significant with p-values smaller than 0.05 in the entire manuscript unless indicated otherwise".
Units in the linear relationship are included in the text.
Correlation equations have been plotted within the figures but without units, since there is insufficient space (Fig. 5, 6 and 11).

*Please have colour blind people in mind for all figures. It is not possible to differentiate between years with the presently used lighter / darker colours; you could use different symbols as well.*

AC: Fig. 5, 6 and 11 have were modified using different symbols.

*Abbreviations that are only used in one paragraph only are superfluous; for example EBUS. Consider abbreviating T by SST and and S by SSS for readability.*

AC: These suggestions have been considered in the text.

*The manuscript will benefit from the input of a native speaker. There is a need to check for incomplete sentences and the use of correct tenses. There should be fewer, but longer paragraphs that consist of more than one or two sentences; while covering the same topic. This will make it easier to follow clear arguments.*

AC: A native speaker with experience of scientific papers has revised the manuscript again.

AC: This final paragraph has been modified to give more clarity. Line 491-494: "The annual uptake capacity of $CO_2$ by the surface waters in our study area is 14.9 Gg C year$^{-1}$. The $CO_2$ fluxes present seasonal variation: these waters act as a source of $CO_2$ to the atmosphere in summer and autumn and as a

375 sink in winter and spring. Based on the information available in the zone, there seems to have been a decrease in the capacity for $CO_2$ capture in the zone in recent decades".

*There is a need to remove some of the tables and figures.*

380 AC: Thank you for your suggestion. Fig. 8 and Table 4 have been removed. Table 5 is now Table 4 and only includes studies in the Gulf of Cádiz. Fig. 1 has been improved.

385

*In summary, this manuscript needs a major revision before publishing.*

AC: Agreed. We hope the substantial revisions we have made to the manuscript now make the paper suitable for publication.

390

References:

Olsen, A., Brown, K. R., Chierici, M., Johannessen, T., Neill, C.: Sea-surface $CO_2$ fugacity in the subpolar North Atlantic, Biogeosciences, 5, 535-547, https://doi.org/10.5194/bg-5-535-2008, 2008.

395

**\* Reviewer 3:**

*The authors investigate factors controlling pCO₂ variations in the Gulf of Cadiz. They use high quality data from 8 cruises incorporating underway data of pCO₂, SSS, SST, and wind speed as well as discrete data for pH, AOU, and nutrients taken along three repeat transects during the cruises. They present spatiotemporal distributions of the underway data, the cruise averages of the discrete data, and the seasonal changes of the computed air-sea CO₂ flux. The authors then discuss the factors influencing the pCO₂ variability. Specifically, they quantify thermal/non-thermal controls of pCO₂. They conclude that temperature and biological activity are the two principal factors that explain the temporal variability of pCO₂. They also point out that continental inputs and mixing with water originating from warm ocean currents influence the spatial variability of pCO₂.*

*The work is OK structured, includes original research based on high quality data, and suits for publication in this journal. However, there are several things that need improvements and/or clarification and I recommend major revision.*

- *General comments*

*1- The main subject of the study is the controls of pCO₂ variations. The authors correctly write "In addition to influence of temperature, the spatiotemporal distribution of pCO₂ in surface seawater is affected by the biological utilization of CO₂, the vertical and lateral transport, the sea-air exchange of CO₂ and terrestrial inputs." However, they do not quantify the relative importance of these controls in their data although there are published methods for such quantification (e.g. Olsen et al 2008). Specifically, the importance of fresh water input and air-sea exchange need to be quantified. This should be feasible since they have seasonal data of two parameters of the CO₂-system in addition to SST, SSS, and nutrients.*

AC: Thank you very much for your suggestion, which is very interesting. The Olsen et al. (2008) method has been taken into account in the revised manuscript and the contributions of SST, air-sea CO₂ exchange and mixing plus biology on pCO₂ change have been quantified. This method also considers change due to SSS variations but we have not included this quantification, since we do not have available the variations of total alkalinity and dissolved inorganic carbon, and the spatial changes of SSS are only significant near the Guadalquivir River mouth (a point included in the text, Line 186-187).
The manuscript has been edited in Material and methods (Line 174-191) and Discussion sections (Line 374-391) to include this quantification. A new figure has been added (Figure 7).

*2- The readability of the manuscript need to be improved. For instance, the study area is quite small, but quite complicated in terms of processes and interactions. Hence, there are a lot of names used in the manuscript (e.g. Gulf of Cadiz Current; AZORES Current; Guadalquivir River; Bay of Cádiz; Cape San Vicente), but locations of these are not shown anywhere in the manuscript. Including these names in the maps/figures would enhance the readability of the manuscript. It is also my opinion that it would be much easier to read the paper if the authors present results in seasonal maps (they do that for CO₂ flux in Fig. 10) and then discuss the controls of pCO₂ changes between seasons and places.*

AC: Thank you for your suggestion. Fig. 1 has been edited to add different processes and names used to improve the readability of the manuscript. However, it is not possible to present results in seasonal maps,except for the spatial distribution of CO₂ fluxes, since for example in Fig. 2, the data interpolated in our database for the same season would not be right.

*Line 19, "On the other side" do you mean "on the other hand?"*

450

AC: Corrected.

*Line 48, after "all other organisms" please add "which increases the concentration of inorganic carbon"*

455 AC: This point has been added in the text, thank you (Line 46).

*Line 50 "generate uncertainty" please replace with "is not clearly defined"*

AC: Corrected.

460

*Lines 62-65 I do not understand the sentences between "Finally, the inner.." and ": : :.towards offshore (Walsh 1991)."*

AC: These sentences have been modified (Line 57-62). The effect of the continental inputs on $pCO_2$
465 variation are explained.

*Line 193, "T values were significantly different among all cruises (p < 0.05)" why is this important result to mention?*

470 AC: Yes, you are correct. In any case, p-values have been removed from the manuscript and a brief reference has been added to the Statistical analysis section.

Line 207-209: "The threshold value for statistical significance was taken as $p < 0.05$. Moreover, all reported linear correlations are type I and they are statistically significant with p-values smaller than 0.05 in the entire manuscript unless indicated otherwise".

475

*Line 96-97 "Spatially T tended to increase from coastal to offshore areas" during all seasons? Or during winter?*

AC: This sentence has been modified in the text. It was not clear in the previous version of the manuscript.
480 Line 217-218: "In general, spatially SST tended to increase from coastal to offshore areas during spring and winter, while in summer and autumn this SST gradient was inverse (Fig. 2A)".

*Lines 211-215. I do not understand. Do you mean that both underway and discrete data are shown in Fig 2B? if so please clarify this in the caption and explain more about the reasons for differences between*
485 *different data.*

AC: Fig. 2B only show the underway measurements, but we have observed an increase of these $pCO_2$
values in the areas that coincide with the position of the discrete water samples (Fig. 1). This sentence has been edited in the manuscript: Line 239-241: "In Fig. 2 a sharp variation of SST and $pCO_2$ can be
490 observed in some zones that coincides with the stations where discrete water samples were taken. This may be due to the different sampling time at these stations, which varied between 2 and 8 hours in function of the depth of the system".

495

*Line 238 "TF presented the highest mean concentration for the whole study (0.77 _ 0.76 _mol L-1)." I notice that given the mean PO4 of 0.28 this mean NO3 is much less than what is expected from Redfield, is this typical for the area?*

AC: Low N/P relationships are typical for this study area (Anfuso et al., 2010). This information has been added in the text: Line 273-275: "The mean N/P ratio in surface waters for the whole study was $3.5 \pm 2.0$, similar to that estimated by Anfuso et al. (2010) in the northeast continental shelf of the Gulf of Cádiz, which indicates a relative phosphate deficit with respect to the Redfield ratio (Redfield et al., 1963)".

*Lines 283-291, please state the uncertainty of the implied $pCO_2$ growth. Please elaborate why you believe the excess $pCO_2$ growth (over the atmospheric growth) is caused by continental input.*

AC: The uncertainty value has been added in the text (Line 309). The $pCO_2$ growth caused by continental inputs was also found by other authors and this point is included in the text.

Line 311-314: "This suggests a possible increase of the anthropogenic nutrient and C inputs from land (Mackenzie et al., 2004) since the direction and magnitude of estuarine and continental shelf $CO_2$ exchange with the atmosphere is highly dependent on the terrestrial organic budget and nutrient supplies to the coastal ocean (Borges and Abril, 2011; Cai, 2011)".

*Lines 300-305, can the reason for difference $pCO_2$ over different depth ranges be due to different TA/DIC ratios in the FW influenced areas and those offshore?*

AC: Thank you for your comment. This is a very interesting question. However, at the moment, we do not have available this information needed to answer it.

*Line 321, in which form is the $CO_2$ input?*

AC: The form of $CO_2$ input is inorganic carbon. This sentence has been edited in the text (Line 331-334).

*Lines 333-334, How $pCO_2$ increase can be computed from only F? or do you make more assumptions?*

AC: More assumptions are necessary: mean depth of the water column (23 m) and the fact that it is well-mixed; a pH = 8; in the conditions of mean temperature and salinity in the Gulf of Cádiz (18.8 °C and 36.19, respectively) and using the K1 and K2 acidity constants proposed by Lueker et al. (2000) in the total pH scale (information indicated in the text: Line 341-346).

*Lines 335-342, you mention that upwelling systems can be influencing the distribution of $pCO_2$ in the Gulf of Cadiz. BUT do you have any evidence for such influence in your data? If not why do you mention it here?*

AC: There is some evidence in our data for the Trafalgar transect. This point is included in the text: Line 351- 354: "In our database experimental evidence of the upwelling was found only in the TF transect. A local decrease of the mean values of SST (17.4 °C) and $pCO_2$ (399.1 µatm) was observed in this coastal area of TF, with respect to the deeper areas (18.8 °C and 405.1 µatm, respectively) for the whole period". In addition, in Fig. 10 it can also observed that the areas near to the Trafalgar section show lower values of $CO_2$ flux during summer and winter.

*Figures: Figure 1: show important currents and places mentioned in the text. Figures 2, 3, 5, 6, and 7. Clarify in the caption whether both underway and discrete data are used.*

AC: Fig. 1 has been improved and the figure captions have been clarified.

References:

[revised manuscript text omitted]
: 19 J̶u̶n̶yJuly 2018). In contrastM̶o̶o̶r̶e̶o̶v̶e̶r̶, the lowest values of $pCO_2$ w̶h̶i̶l̶e̶ ̶i̶n̶ were recorded in t̶h̶e̶ spring of 2015 (̶S̶T̶5̶,̶ ̶d̶a̶r̶k̶ ̶g̶r̶e̶e̶n̶)̶ ̶t̶h̶e̶ ̶l̶o̶w̶e̶s̶t̶ ̶v̶a̶l̶u̶e̶s̶ o̶f̶ ̶t̶h̶e̶ ̶s̶t̶u̶d̶y̶ ̶w̶e̶r̶e̶ ̶r̶e̶c̶o̶r̶d̶e̶d̶ in this zone (as low as 320 µatm) d̶u̶r̶i̶n̶gin a period of drought (flow rate 20 $m^3 s^{-1}$) and subject to

1010 intense biological activity associated with the highest value found of the concentration of chlorophyll-a (2.4 $µg L^{-1}$). The Bay of Cádiz occupies an area of 38 $km^2$, and receives urban effluents from a population of 640,000 inhabitants. This shallow zone is oversaturated with $CO_2$ (Ribas-Ribas et al., 2011) due largely to the inputs of inorganic carbon, organic matter and nutrients that are received from the Guadalete River and Sancti Petri Channel and the Río San Pedro tidal creeks (de la Paz et al., 2008a, b; Burgos et al, 2018).

1015 .

1020 Another source of $CO_2$ in the coastal zone results from the net production of inorganic carbon derived from the processes of remineralization of the organic matter in the surface sediments originat̶i̶n̶ged from the continuous deposition of organic matter through the water column (de Haas et al., 2002; Jahnke et al., 2005). The intensity of this process decreases in line with the increasing depth of the system, and the influence of the primary production and the continental supplies on the deposition of the particulate organic matter is less (Friedl et al., 1998; Burdige, 2007; Al Azhar et al., 2017). Ferrón et al. (2009) quantified

1025 the release from the sediment of DIC related to the processes of oxidation of organic matter in the coastal zone (depth < 50 m) of the Gulf of Cádiz, between the Guadalquivir and the Bay of Cádiz. These authors found a mean benthic flux of 27 ± 8 mmol C $m^{-2} d^{-1}$ for stations with a mean depth of 23 m. C̶o̶n̶s̶i̶d̶e̶r̶i̶n̶g̶ ̶a̶ ̶w̶e̶l̶l̶-̶m̶i̶x̶e̶d̶ ̶w̶a̶t̶e̶r̶ ̶c̶o̶l̶u̶m̶n̶,̶ ̶a̶ ̶p̶H̶ ̶=̶ ̶8̶,̶ ̶i̶n̶ ̶t̶h̶e̶ ̶c̶o̶n̶d̶i̶t̶i̶o̶n̶s̶ ̶o̶f̶ ̶m̶e̶a̶n̶ t̶e̶m̶p̶e̶r̶a̶t̶u̶r̶e̶ ̶a̶n̶d̶ ̶s̶a̶l̶i̶n̶i̶t̶y̶ ̶i̶n̶ ̶t̶h̶e̶ ̶G̶u̶l̶f̶ ̶o̶f̶ ̶C̶á̶d̶i̶z̶ ̶(̶1̶8̶.̶8̶ ̶°̶C̶ ̶a̶n̶d̶ ̶3̶6̶.̶1̶9̶,̶ ̶r̶e̶s̶p̶e̶c̶t̶i̶v̶e̶l̶y̶)̶ ̶a̶n̶d̶ ̶u̶s̶i̶n̶g̶ ̶t̶h̶e̶ ̶K̶1̶ ̶a̶n̶d̶ ̶K̶2̶ ̶a̶c̶i̶d̶i̶t̶y̶ ̶c̶o̶n̶s̶t̶a̶n̶t̶s̶ p̶r̶o̶p̶o̶s̶e̶d̶ ̶b̶y̶ ̶L̶u̶e̶k̶e̶r̶ ̶e̶t̶ ̶a̶l̶.̶ ̶(̶2̶0̶0̶0̶)̶ ̶i̶n̶ ̶t̶h̶e̶ ̶t̶o̶t̶a̶l̶ ̶p̶H̶ ̶s̶c̶a̶l̶e̶.̶ 
[revised manuscript text omitted]

---

## Referee Report (RR1)

Review on "pCO$_2$ variability in the surface waters of the eastern Gulf of Cádiz (SW Iberian Península)" by Jiménez-López et al.

General comments

The authors use the approach by Olsen et. al. (2008) that allows to quantify different contributions to the total observed change in pCO$_2$, yet the authors fail to implement this in their discussion. First, complex local interactions are described without using the evidence found in the data (Line 316-373), before the quantification of different contributions appears - more as an after-thought (Line 374-391). Moreover, the authors now describe two different ways to estimate the pCO$_2$ decomposition in their method section, the original Takahashi approach and the more elaborate decomposition as in e.g. Olsen et. al. (2008). There is no fundamental difference between Equation (1) and (7), but the authors treat it as such in the method and discussion section. The authors should restructure and revise both sections accordingly.

The authors should explain, how the uncertainty of the measurements is determined - is it a standard deviation in space or time? Many reported values lack statistical significance to support the statements made by the authors (probably a consequence of averaging over the entire study area and all seasons), which is not acknowledged or discussed or explained throughout the manuscript.

The manuscript has improved since the first submission, but there are still issues that need to be addressed before the manuscript is ready for publication.

Minor comments

Line 152-168: Again, paraphrase the method by Takahashi et al. 2002. At the moment, many sentences are directly copied from the original paper without citation.

Line 162-165: You sample 4 times a year and according to Figure 3, you do not sample the maximum SST in summer, therefore you do not capture the true seasonal amplitude of SST (or pCO$_2$). You should explain that you estimate here the differences between summer and winter cruises.

Line 177: superscript SW needs to be explained

Line 186-188: The residual may be dominated by mixing and biological activity, but it also includes salinity-driven and freshwater-induced changes in pCO$_2$ and other minor processes that impact surface pCO$_2$. In any case, I do not understand, why the salinity-driven and freshwater-driven changes of pCO$_2$ are not

calculated as through the presence of large river system this may not be negligible. Moreover, two parameters of the $CO_2$ system in seawater are measured, that is sufficient to estimate DIC and alkalinity. Follow e.g. Sarmiento and Gruber (2006).

Line 236-238: The reported values do not support this statement; the values are not statistically significant different from zero, except for the winter value.

Line 242-247: Again, why is Figure 4 helpful: there is no statistical difference in both SST and $pCO_2$ with bottom depth range; there is no general trend to be observed here. There is neither a decrease nor a progressive increase to be observed in the mean values, but there might be, if you look at seasonal values.

Line 281-283: The reported values do not support this statement; only $CO_2$ fluxes during ST3 and ST5 are statistically significant different from zero; only autumn is statistically significant different from zero.

Line 324: Is C dissolved inorganic or organic carbon? abbreviation without explanation.

Line 351-352: same sentence.

Line 355-357: The present Figure 4 does not support this statement.

Line 374: Figure 7? please recheck all Figure references in the manuscript!

Line 375: Again, the residual also represents salinity-driven and freshwater-induced changes in $pCO_2$

Line 377: "[...] presents practically the same temporal trend in deep and coastal areas, but with a global behaviour different [...]" - what do you mean by that?

Line 379: The reported values do not support this statement; neither is the distal zone a sink nor the coastal area a source of $CO_2$ as both values are not statistically significant different from zero.

Line 374-391: Consider discussing $dpCO_2/dt$ instead. Between cruises not the same amount of time has passed and therefore Figure 7 includes a temporal bias that has no physical reason.

Line 449: The reported values do not support this statement; both values are not statistically significant different from each other.

Line 459-460: The reported values do not support this statement; both values are not statistically significant different from zero.

Line 465: The reported value does not support this statement; it is not statistically significant different from zero.

Line 467: Please re-check your estimate of the uptake capacity and correct - if necessary - in the entire manuscript:
$0.07 \, \mathrm{mol\,C\,m^{-2}\,yr^{-1}} * 5.28{*}10^9 \, \mathrm{m^2} * 12.01 \, \mathrm{g\,mol^{-1}} = 4.44{*}10^9 \, \mathrm{g\,C\,yr^{-1}}$

Line 473: The reported values does not support this statement; both values are not statistically significant different from each other.

Line 493-493: The statement made in the last sentence has not been discussed throughout the paper and needs further explanation.

Figure 10 and 11: Please indicate: air-sea or sea-air $CO_2$ flux

References:

Sarmiento, J. L. and Gruber, N.: Ocean Biogeochemical Dynamics, Princeton University Press, Princeton, New Jersey, USA, 2006.

---

## Author Response (AR2)

Dear Editor,

The authors are very grateful with the second revision of the manuscript by two of the anonymous reviewers. We have addressed the reviewers' annotations and revised carefully the manuscript in order to improve our work again.

Below you will find the comments they made and our comments as authors (marked AC) on each point. In response to all the comments the manuscript has been modified, resulting in changes to line numbers. Therefore, we have included the new line numbers (whenever applicable) so that you can refer to either the current or (former) version if you wish.

Thank you very much for your consideration.

Sincerely,

Dolores Jiménez-López, on behalf of all co-authors.

**\* Reviewer 1:**

- *General comments*

*The authors use the approach by Olsen et. al. (2008) that allows to quantify different contributions to the total observed change in pCO$_2$, yet the authors fail to implement this in their discussion. First, complex local interactions are described without using the evidence found in the data (Line 316-373), before the quantification of different contributions appears - more as an after-thought (Line 374-391). Moreover, the authors now describe two different ways to estimate the pCO$_2$ decomposition in their method section, the original Takahashi approach and the more elaborate decomposition as in e.g. Olsen et. al. (2008). There is no fundamental difference between Equation (1) and (7), but the authors treat it as such in the method and discussion section. The authors should restructure and revise both sections accordingly.*

AC: Thank you very much for this suggestion. We have modified this section of the manuscript (Section 4.2, now Line 330-411).
The section has been restructured, first the quantification of different processes was described (Line 331-352) and then it has been related with local interactions found in the Gulf of Cádiz (Line 353-411). Moreover, in the application of the Olsen et al. (2008) method, the pCO$_2$ change brought about by SSS has been included. This has caused modifications in different sections of the manuscript, in Material and Methods (Line 191-194), in Discussion (Line 349-352) and in the Figure 6.
We consider that differences between Equation (1) and (7) exist, although both are based in Takahashi et al. (2002) method. But they are used for aims different, since with the Equation (7), we obtain pCO$_2$ variations between cruises.

*The authors should explain, how the uncertainty of the measurements is determined - is it a standard deviation in space or time? Many reported values lack statistical significance to support the statements made by the authors (probably a consequence of averaging over the entire study area and all seasons), which is not acknowledged or discussed or explained throughout the manuscript.*

AC: Thank you, we agree with this suggestion. We have tried to explain the uncertainty of the measurements in different paragraphs in the entire manuscript since the work presents a high spatial and seasonal variability and also standard deviations values depend on the focus considered along our work. For example, with the following tables can be observed the variability of SST, SSS and pCO$_2$ in the Gulf of Cádiz for two focus different.

Table 1: Average values and standard deviation of underway SST, SSS and pCO$_2$ during the 8 cruises undertaken.

| Cruise | SST (ºC) | SSS | pCO$_2$ (µatm) |
|--------|----------|-----|----------------|
| ST1 | 15.4 ± 0.6 | 36.11 ± 0.18 | 396.5 ± 19.0 |
| ST2 | 21.1 ± 0.9 | 36.21 ± 0.15 | 412.9 ± 12.6 |
| ST3 | 21.5 ± 1.3 | 36.26 ± 0.22 | 413.5 ± 9.8 |
| ST4 | 18.1 ± 0.7 | 36.36 ± 0.21 | 388.7 ± 12.9 |
| ST5 | 15.6 ± 0.4 | 36.12 ± 0.14 | 368.6 ± 14.9 |
| ST6 | 20.9 ± 0.8 | 36.40 ± 0.08 | 410.3 ± 13.8 |
| ST7 | 20.6 ± 1.1 | 35.64 ± 0.08 | 407.6 ± 11.2 |
| ST8 | 16.8 ± 0.4 | 36.44 ± 0.09 | 392.9 ± 17.9 |

Table 2: Average values and standard deviation of underway SST, SSS and $pCO_2$ during the 8 cruises undertaken in function of different bottom-depth ranges.

| Bottom-depth ranges (m) | SST (ºC) | SSS | $pCO_2$ (µatm) |
|---|---|---|---|
| < 50 m | 18.4 ± 2.8 | 36.09 ± 0.29 | 408.3 ± 26.7 |
| 50 - 100 m | 18.3 ± 2.6 | 36.11 ± 0.24 | 396.1 ± 23.0 |
| 100 - 200 m | 18.6 ± 2.5 | 36.16 ± 0.25 | 396.0 ± 19.0 |
| 200 - 400 m | 19.1 ± 2.5 | 36.23 ± 0.25 | 397.4 ± 17.1 |
| 400 - 600 m | 19.4 ± 2.4 | 36.22 ± 0.25 | 400.1 ± 18.3 |
| > 600 m | 20.1 ± 2.4 | 36.34 ± 0.28 | 404.3 ± 16.5 |

In the previous revision, the section 2.5 related with the Statistical analysis was modified to try to simplify the statistical significant differences found in the entire manuscript (Line 214-215) unless indicated otherwise.

*The manuscript has improved since the first submission, but there are still issues that need to be addressed before the manuscript is ready for publication.*

- *Minor comments*

*Line 152-168: Again, paraphrase the method by Takahashi et al. 2002. At the moment, many sentences are directly copied from the original paper without citation.*

AC: The description of the method described by Takahashi et al. (2002) has been modified in this section of the manuscript (now Line 152-176).

*Line 162-165: You sample 4 times a year and according to Figure 3, you do not sample the maximum SST in summer, therefore you do not capture the true seasonal amplitude of SST (or $pCO_2$). You should explain that you estimate here the differences between summer and winter cruises.*

AC: It is true that we do not sample the full seasonal amplitude of the study area and we know that the range in the $pCO_2$ variation could be greater than the determined in this work, as it has been included in the text (Line 234-236). We use "seasonal amplitude" of SST or $pCO_2$ in Material and Methods section to refer it to our sampling, although the maximum SST during August in the Gulf of Cádiz are not registered.

*Line 177: superscript SW needs to be explained.*

AC: Corrected. The superscript has been explained in the line 181 ("…"sw" makes reference to the surface $pCO_2$ in the seawater…").

*Line 186-188: The residual may be dominated by mixing and biological activity, but it also includes salinity-driven and freshwater-induced changes in $pCO_2$ and other minor processes that impact surface $pCO_2$. In any case, I do not understand, why the salinity-driven and freshwater-driven changes of $pCO_2$ are not calculated as through the presence of large river system this may not be negligible. Moreover, two parameters of the $CO_2$ system in seawater are measured, that is sufficient to estimate DIC and alkalinity. Follow e.g. Sarmiento and Gruber (2006).*

AC: Thank you very much for your suggestion. We have included the salinity-driven and freshwater-induced changes about $pCO_2$, although its effect is very low compared with other processes described (Line 349-352 and new Figure 6).
In the Gulf of Cádiz, the $pCO_2$ changes brought about by SSS provide an increase of $pCO_2$ of 15 µatm approximately by 1 unit of variation of salinity. But in this work, we have estimated with the Olsen et al. (2008) method that the contribution of this effect is lowest than other processes, since the SSS changes are low between cruises in our study area (< 0.8 units).
The two parameters necessary to estimate this effect, DIC and TA, have been estimated with the program CO2SYS through pH and $pCO_2$ values (indicated in the manuscript, Line 192-194). Although the relationships between SSS with TA and DIC generate a certain degree of uncertainty (e.g. for the coastal zones, $r^2=0.36$ and $r^2=0.33$, respectively).

*Line 236-238: The reported values do not support this statement; the values are not statistically significant different from zero, except for the winter value.*

AC: We agree and a little reference has been included in the text (Line 245-246).

*Line 242-247: Again, why is Figure 4 helpful: there is no statistical difference in both SST and $pCO_2$ with bottom depth range; there is no general trend to be observed here. There is neither a decrease nor a progressive increase to be observed in the mean values, but there might be, if you look at seasonal values.*

AC: Yes, we agree with your suggestion, although the only objective of this figure is to show the variation of the mean values of $pCO_2$ and SST between coastal and deep areas.
We know that there are not statistical significant differences in both SST and $pCO_2$ with bottom-depth ranges (see Table 2 of the General comments) and this has been clarified in the text (Line 255-256). Although with this Figure 4, we only want to show the general trend of these variables with the depth of the system. Maybe the representation of the variations of SST and $pCO_2$ for the different transects can better show this trend but this way appears us little useful and less representative for the entire work.

*Line 281-283: The reported values do not support this statement; only $CO_2$ fluxes during ST3 and ST5 are statistically significant different from zero; only autumn is statistically significant different from zero.*

AC: Suggestion added and clarified in the manuscript (Line 293-295).

*Line 324: Is C dissolved inorganic or organic carbon? abbreviation without explanation.*

AC: The abbreviation of C has been specified (now Line 388).

*Line 351-352: same sentence.*

AC: Corrected.

*Line 355-357: The present Figure 4 does not support this statement.*

AC: Although there are not statistical significant differences, we observe a little increase of the mean values of SST and $pCO_2$ with the depth of the system associated with the presence of a warmer branch of the Azores Current.

*Line 374: Figure 7? please recheck all Figure references in the manuscript!*

AC: Our apologies, this citation has been corrected. Also the Figure 7 has been modified and now is the Figure 6.

*Line 375: Again, the residual also represents salinity-driven and freshwater-induced changes in $pCO_2$.*

AC: Modified. This effect has been included in the application of the Olsen et al. (2008) method (Line 349-352).

*Line 377: "[...] presents practically the same temporal trend in deep and coastal areas, but with a global behaviour different [...]" - what do you mean by that?*

AC: This sentence has been clarified in the manuscript (Line 335-338). $pCO_2$ variation in the sea surface water presents a similar variation in coastal and deep areas, although the processes that affect to the $pCO_2$ changes are different in each area.

*Line 379: The reported values do not support this statement; neither is the distal zone a sink nor the coastal area a source of $CO_2$ as both values are not statistically significant different from zero.*

AC: This sentence has been clarified in the text (Line 338).

*Line 374-391: Consider discussing $dpCO_2/dt$ instead. Between cruises not the same amount of time has passed and therefore Figure 7 includes a temporal bias that has no physical reason.*

AC: The time between cruises is similar in all the sampling, and this fact has been added in the manuscript.

Line 334-335: "The average time between cruises is $86 \pm 8$ days, with the exception of the last period (between September 2015 and February 2016) that was 140 days".

*Line 449: The reported values do not support this statement; both values are not statistically significant different from each other.*

AC: We agree with your suggestion but there is a high seasonal variability in our work. This sentence was clarified in the manuscript (Line 471-472).

*Line 459-460: The reported values do not support this statement; both values are not statistically significant different from zero.*

AC: Sentence added and clarified in the text (Line 483-484).

*Line 465: The reported value does not support this statement; it is not statistically significant different from zero.*

AC: We agree and we have added a sentence to clarify this statement in the manuscript.

Line 489-490: "The Gulf of Cádiz, during the period of this sampling, could act as a sink of $CO_2$, with a mean rate of -0.18 ± 1.32 mmol m$^{-2}$ d$^{-1}$, even though it is necessary to consider the intrinsic variability of the database that generate a high standard deviation".

*Line 467: Please re-check your estimate of the uptake capacity and correct – if necessary - in the entire manuscript:*
*0.07 molCm$^{-2}$ yr$^{-1}$ * 5.28*10$^9$ m$^2$ * 12.01 g mol$^{-1}$ = 4.44*10$^9$ gCyr$^{-1}$*

AC: Our apologies, the correct value is 4.1 Gg C year$^{-1}$. This has been rectified in the manuscript (Line 492) and in the entire manuscrpit. Thank you for drawing this to our attention.

*Line 473: The reported values does not support this statement; both values are not statistically significant different from each other.*

AC: Yes, we know that the values do not support this statement and for this we have added a sentence to clarify it in the manuscript.

Line 498-499: "pCO$_2$ database presents a high variability in the Gulf of Cádiz associated with its location, it is a transition zone between coastal and shelf area, and the own seasonal variation".

*Line 493-493: The statement made in the last sentence has not been discussed throughout the paper and needs further explanation.*

AC: This statement has been better explained in the manuscript.

Line 519-521: "Based on the information available in the zone, there seems to have been a decrease in the capacity for $CO_2$ capture in the zone in recent decades, since the pCO$_2$ has increased from 360.6 ± 18.2 µatm (Ribas-Ribas et al., 2011) to 398.9 ± 15.5 µatm in the actuality".

*Figure 10 and 11: Please indicate: air-sea or sea-air CO$_2$ flux*

AC: The captions of these figures were modified.

 **\* Reviewer 3:**

*I find the author response to my comments (and major comments from other referees) adequate. Hence, I recommend publication after the following minor improvements of the revised manuscript.*

*Line 69, remove "global".*

235

AC: Corrected.

*Section 4.1: the summary of earlier results on Table 4 is very useful for the readability of the text. Throughout the discussion the authors cite values from these earlier studies (e.g. mean $pCO_2$ values*
240 *(line 304); $CO_2$ flux values (line 436-437); T/B ratio (line 393-397)). However, Table 4 shows only $pCO_2$ ranges and $FCO_2$ values. I suggest to enter $pCO_2$ mean values and T/B ratios in Table 4. Note also that the caption of Table 4 says "Mean and ranges…" although only the ranges are shown.*

AC: $pCO_2$ mean values and T/B ratios were added in the Table 4 and the caption has been modified,
245 thank you.

*Line 325 (and elsewhere applicable), replace "C" with "carbon" – if you know the for please specify e.g. "inorganic carbon".*

250 AC: Corrected in the line 325 (now Line 388), and it was also modified in the line 312 (now Line 327).

*The paragraph in lines 404-419 can be improved by:*
*-extending the first sentence with e.g. "which is observed to decrease from coast to deeper zone regardless which method is used (Takahashi et al., XXXX normalization method or decomposition*
255 *method of Olsen et al 2008)."*
*-removing the last sentence*
*-removing "thus" in line 412 and "however" in line 415.*

AC: Corrected, thank you for your suggestion.
260

*Line 475, replace "global" with an alternative phrase to avoid possible confusion.*

AC: Global has been replaced and this sentence was modified.

[revised manuscript text omitted]

---

## Author Response (AR3)

Dear Editor:

The authors are very grateful with your revision. In the new revised version of the manuscript, all the suggestions and comments have been taken into account. Some suggestions have also been clarified and changed wording in the text.

Below you will find the comments you made and our comments as authors (marked AC) on each point. In response to all your comments the manuscript has been modified, resulting in some changes to line numbers. Therefore, we have included the new line numbers (whenever applicable) so that you can refer to either the current or (former) version if you wish.

Thank you very much for your consideration.

Sincerely,

Dolores Jiménez-López, on behalf of all co-authors.

15  *Topic Editor Decision: Publish subject to minor revisions (review by editor) (06 Aug 2019) by Mario Hoppema*

*Comments to the Author:*

*Dear Dr. Jiménez-López and co-authors,*

*Thanks for your revised manuscript, which I have read and corrected. My comments and listed below.*
20  *Your manuscript is almost there. Please prepare another revised version taking into account the below comments:*

*L28 de Haas (typo)*

AC: Corrected.

25  *L29 I suggest: ... total carbon sequestration through the mechanism of the biological pump ...*

AC: Changed.

*L33 ... lower than that ...*

AC: Modified.

*L42 I suggest: At the continental shelf a high spatiotemporal variability of the air-sea $CO_2$ fluxes occurs*
30  *due to ...*

AC: Modified.

*L42-43 "upwelling zones" and "input" are not processes. Please change wording.*

AC: The wording has been changed.

Line 42-43: "At the continental shelf, a high spatiotemporal variability of the air-sea $CO_2$ fluxes occurs
35  due to various effects, such as the thermodynamic effects, the biological processes, the gas exchange, the upwelling zones and the continental inputs".

*L58 "more affected" More than what?*

AC: Included.

Line 59: "… than the outer shelf".

40  *L62 shows (instead of: presents)*

AC: Changed.

*L63 "in this zone" This formulation is not clear; is this the former or the latter? Please make clear*

AC: Clarified in the text.

Line 63: "… in the inner zone".

45  *L64 "The Gulf of Cádiz is a geographical domain of considerable interest due to its location." This is a strange formulation. I guess it holds for every oceanic region. Please specify and change wording.*

AC: The wording has been changed.

Line 65-67: "The Gulf of Cádiz is strategically located connecting the Atlantic Ocean with the Mediterranean Sea through the Strait of Gibraltar and in addition it receives continental inputs from several major rivers, i.e. the Guadalquivir, Tinto, Odiel and Guadiana".

*L133 Please define CTD here*

AC: Defined and included.

Line 133: "…SeaBird CTD 911+ (Conductivity-Temperature-Depth system)".

*L148 Please use SI units instead of mile*

AC: Modified and included.

Line 149: "…to 0.9 km around the location of the fixed stations approximately".

*L196 please add "water depth" to the > 50 m and < 50 m, as this is not clear now*

AC: Added in the text.

Line 199-200: "…between coastal (water depth < 50 m) and distal (water depth > 50 m) areas".

*L212 add USA after NY (and write full New York, correct?)*

AC: You are right, NY is New York. Modified and added USA in the text.

Line 215: "Statistical analyses were performed with IBM SPSS Statistics software (Version 20.0. Armonk, New York, USA)".

*L219 delete: sampling*

AC: Deleted (Line 221).

*L221-222 I suggest: During 2014, SST values were found to be higher than those in 2015 and 2016*

AC: Changed (Line 224).

*L222 delete: for both seasons*

AC: Deleted (Line 225).

*L224 delete: spatially*

AC: Deleted (Line 226).

*L234-236 This sentence is not clear; I suggest: Sampling during our cruises did not detect the highest temperatures occurring in the Gulf of Cádiz during August, which may indicate that the real range of pCO$_2$ variation be greater than that determined in this study.*

AC: Modified (Line 237-239).

*L237 Change to: The highest mean values … (since in the previous sentence a higher value was given)*

AC: Changed (Line 240).

*L245-246 This sentence is not clear; change to: These mean values are not significantly different and the standard deviations are high, indicating high spatial and inter-annual variability.*

AC: Modified (Line 248-249).

*L273 variable (instead of: parameter)*

AC: Modified (Line 274).

*L294 ... in the CO$_2$ fluxes ... (delete: dataset of)*

AC: Deleted (Line 295).

*L294 ... similar to pCO$_2$ ... (not: as in the values of pCO$_2$)*

AC: Changed (Line 295).

*L305-306 The sentence is unclear. I suggest: In our study, all data from all seasons together exhibited a linear relationship between pCO$_2$ and SST (r2 = 0.37, Fig. 5A).*

AC: Sentence changed (Line 305-306).

*L306 becomes even more significant ... (add: even)*

AC: Added (Line 306).

*L326-327 Is there any additional evidence, or only indications, from research on land that such is happening in this region?*

AC: We believe that this effect could be one of the causes of the increase of pCO$_2$ in the surface waters of the Gulf of Cádiz. But for now we do not have any additional evidence to confirm it. This sentence has been modified and added this information in the text.

Line 326-330: "The cause of this increase could be a greater input of anthropogenic nutrients and inorganic carbon from land (Mackenzie et al., 2004) since the direction and magnitude of estuarine and continental shelf CO$_2$ exchange with the atmosphere is highly dependent on the terrestrial organic budget and nutrient supplies to the coastal ocean (Borges and Abril, 2011; Cai, 2011). Although we do not have any additional evidence to confirm this effect in our area of study currently".

*L337 "the shallower areas as a source of CO$_2$ (mean = 0.2 ± 22.7" I wouldn't call that a source. This is clearly neither source nor sink, i.e., neutral.*

AC: Sentence modified in the text.

Line 335-337: "$dpCO_2^{sw}$ presents a similar variation between deep and coastal areas, but with small differences in the mean values between the distal zones ($dpCO_2^{sw}$ = -3.4 ± 28.9 µatm) and the shallower areas ($dpCO_2^{sw}$ = 0.2 ± 22.7 µatm)".

*L340 biological processes*

AC: Modified (Line 340).

*L342-343 ... the changes produced by air-sea CO$_2$ exchange are relatively small.*

AC: Modified (Line 343).

*L344-345 I suggest to change this to: In regions with shallower mixed layers, the effect of air-sea exchange on the pCO$_2$ variation is larger (Olsen et al., 2008).*

AC: Changed (Line 345-346).

*L351-352 ... except for an area influenced by continental runoff where pCO₂ decreases.*

AC: Modified (Line 352-353).

*L353 biological processes*

AC: Modified (Line 354).

120 *L353-355 ... a dependence between the mean values of pCO₂ and pH, AOU and the concentration of chlorophyll-a has been observed at the fixed stations (n = 126, Fig. 7)*

AC: Changed (Line 354-356).

*L355 AOU and pCO₂ show a positive relationship ...*

AC: Modified (Line 356).

125 *L358 delete: However, (this relationship is expected, no surprise)*

AC: Deleted (Line 359-360).

*L366-367 I suggest to change to: Something that could affect the distribution of pCO₂ (and be considered to be part of mixing and biology sensu Olsen et al., 2008), is vertical and lateral transport.*

AC: This suggestion has been accepted and modified in the text (Line 367-368).

130 *L380 delete: Additionally,*

*Or for that reason delete the whole sentence in L380-381: "Additionally, several authors have described the influence of the continental inputs on the distribution of pCO₂ in surface waters." (because the authors are mentioned in L383-384)*

AC: The wording has been changed in the text.

135 Line 381-382: "Additionally, related with the lateral transport on the distribution of pCO₂ in surface waters, several authors have described the influence of the continental inputs".

*L380-384 Why is this, what you describe? It would be interesting to read the reason here.*

AC: We describe the effect of the lateral transport on the distribution of pCO₂. For this reason, the wording has been clarified in the text, as you can see in the previous point.

140 *L385 derive (instead of: take place)*

AC: Modified (Line 387).

*L402-405 It is not clear to me what you mean here. Please modify sentence.*

AC: This sentence has been clarified in the text.

Line 403-406: "The intensity of this effect decreases towards offshore areas, since the influence of the
145 primary production and the continental supplies on the deposition of the particulate organic matter are less (Friedl et al., 1998; Burdige, 2007; Al Azhar et al., 2017), which could be related with the greater effect determined by the mixing and biology processes in the coastal areas using the Olsen et al. (2008) method".

150     *L407-408 "These authors found a mean benthic flux of $27 \pm 8$ mmol C $m^{-2}$ $d^{-1}$ for stations with a mean depth of 23 m. This flux of DIC is equivalent to a $CO_2$ flux of $198 \pm 80$ µmol C $m^{-2}$ d, considering a well-mixed water column, a pH = 8 ..." I do not understand this calculation. A benthic C flux is given, and from this another C flux is calculated with quite a different amount of C involved. Is the derived flux over a different interface? Please explain what you mean and how you did the calculation.*

155     AC: We calculate the $CO_2$ benthic flux ($198 \pm 80$ µmol C $m^{-2}$ d) through the CID benthic flux ($27 \pm 8$ mmol C $m^{-2}$ $d^{-1}$) with the program CO2SYS (Lewis et al., 1998) but in the same interface. For this, the variables indicated in the text are necessary; the pH (8), the mean temperature and salinity in the Gulf of Cádiz (18.8 ºC and 36.19, respectively) and the K1 and K2 acidity constants proposed by Lueker et al. (2000) in the total pH scale that they are chosen in the own program. And to estimate the influence of

160     this $CO_2$ benthic flux about the $pCO_2$ values in the water column, it is also necessary to considerate the depth of this water column (23m).

This sentence has been clarified in the manuscript. Line 409-414: "This flux of DIC is equivalent to a $CO_2$ flux of $198 \pm 80$ µmol C $m^{-2}$ $d^{-1}$ through the sediment-water interface, considering a well-mixed water column, a pH = 8, in the conditions of mean temperature and salinity in the Gulf of Cádiz (18.8

165     ºC and 36.19, respectively) and using the K1 and K2 acidity constants proposed by Lueker et al. (2000) in the total pH scale through the program CO2SYS (Lewis et al., 1998). Moreover, this estimated $CO_2$ benthic flux would produce an increase of $pCO_2$ of $0.25 \pm 0.10$ µatm $d^{-1}$ in the water column".

*L425 Please add here what high values (of which variable?)*

AC: Variable added in the text ($\Delta pCO_2$ non-thermal) (Line 428-429).

170     *L440-441 delete: that have been cruised for sampling in the study zone (not necessary, the transects have been introduced earlier in the text)*

AC: Deleted (Line 443).

*L441 delete: It can be appreciated that*

AC: Deleted (Line 443).

175     *L445 upwelling (no –s)*

AC: Modified (Line 447).

*L447 show (not: present)*

AC: Modified (Line 449).

*L457 the air-sea flux ...*

180     AC: Added in the text (Line 458).

*L457 shows, or exhibits (not: presents)*

AC: Changed (Line 458).

*L458-459 I suggest to change to: As can be seen in Fig. 10, seasonal and spatial variations were observed in the air-sea fluxes during the period studied.*

185     AC: This sentence has been modified (Line 459-460).*L463 As discussed above for $pCO_2$, temperature change is ...*

AC: Modified (Line 464).

*L466 as a consequence of the biological utilization of the $CO_2$ and the subsequent tendency for $CO_2$ undersaturation … (please note, biological utilization as such does not cause air-sea fluxes)*

AC: Modified (Line 467-468).

*L467 Such relationships*

AC: Changed (Line 468).

*In §4.4 several times $CO_2$ fluxes are mentioned, where air-sea fluxes are meant. Please insert air-sea in some cases for clarity.*

AC: All the terms of "air-sea" have been added in this section. Thank you for your suggestion.

*L489 during the sampling period*

AC: Modified (Line 491).

*L489 With the number of -0.18 mmol $m^{-2}$ $d^{-1}$ including the order of magnitude larger standard deviation it is not very convincing to call the region a $CO_2$ sink. Please change the wording, and consider to use neither source nor sink.*

AC: This sentence has been modified in the manuscript.

Line 491-492: "The Gulf of Cádiz, during the sampling period, shows a mean rate of -0.18 ± 1.32 mmol $m^{-2}$ $d^{-1}$, even though it is necessary to consider the intrinsic variability of the database that generate a high standard deviation".

*L498-499 My suggestion for this sentence: A high variability in $pCO_2$ in the Gulf of Cádiz was observed which is associated with its location as a transition zone between coastal and shelf areas, superimposed on the usual seasonal variation due to thermal and biological effects.*

AC: This sentence has been modified in the text with your suggestion (Line 499-500).

*L499-500 "The mean value of $pCO_2$ found in this study (398.9 ± 15.5 µatm) indicates that the eastern part of the Gulf of Cádiz could be slightly undersaturated in $CO_2$ with respect to …" This is not sound. A mean value for the entire region is given, which you couple to (only) the eastern part of the Gulf that may be undersaturated. Please correct, or explain and modify the text.*

AC: Sentence modified in the manuscript. We refer to the Gulf of Cádiz now.

Line 500-502: "The mean value of $pCO_2$ found in this study (398.9 ± 15.5 µatm) indicates that the Gulf of Cádiz could be slightly undersaturated in $CO_2$ with respect to the atmosphere (402.1 ± 3.9 µatm)".

*L502 littoral (typo)*

AC: Corrected (Line 503).

*L502-503 delete: in this work*

AC: Deleted (Line 503).

*L503-504 biological activity, represented by chlorophyll-a*

AC: Added in the text (Line 505).

*L510 The mean T/B ratio (1.15) of the region suggests …*

225 AC: Modified (Line 512). But in our work, we calculate the total T/B ratio, it is not mean T/B ratio, because we obtain a T/B ratio for the 8 cruises.

*L510 … by temperature changes.*

AC: Modified (Line 512).

*L510-512 I suggest the following: However, the situation is more complicated if the ratio is considered*
230 *as a function of bottom depth, which is associated with the existence of non-thermal processes.*

AC: Modified with your suggestion (Line 513-514).

*L515-516 "In contrast, the highest T/B ratio values have been found in the SP transect, where values of up to 1.54 are obtained for depths greater than 100 m." With those explanations for the lower ratio earlier, what is the reason for this higher ratio? Please add some text on this.*

235 AC: This sentence has been clarified in the manuscript.

Line 517-519: "In contrast, the highest T/B ratio values have been found in the SP transect, where values of up to 1.54 are obtained for depths greater than 100 m, probably related to the greater effect of thermal processes".

*L520-521 "… since the $pCO_2$ has increased from 360.6 ± 18.2 μatm (Ribas-Ribas et al., 2011) to 398.9*
240 *± 520 15.5 μatm in the actuality." It is important to provide the years for these values and the accompanying increase of $pCO_2$ in the atmosphere to be able to judge the contention. Please add text.*

AC: The years have added and the increase of $pCO_2$ in the atmosphere was included in the Line 325 of the manuscript, although we have also added in this section.

Line 523-525: "…since the $pCO_2$ has increased from 360.6 ± 18.2 μatm in a study realised between
245 2006 and 2007 (Ribas-Ribas et al., 2011) to 398.9 ± 15.5 μatm in the actuality and this exceeds the rate of increase of $pCO_2$ in the atmosphere (2.3 μatm year$^{-1}$ for the last 10 years)".

*L905 suggest to write here: data from González-García et al. (2018).*

AC: Added (Line 906).

*title Table 3: mention that this concerns: air-sea $CO_2$ fluxes*

250 AC: Modified.

*title Table 4: idem ditto*

AC: Modified.

*L939 caption Figure 2: please add units of variables shown*

AC: Variables added in the figure caption.

255 *L989 caption Figure 6: Observed changes in $pCO_2$ (first row) and $pCO_2$ changes broken down due to:*

AC: Caption modified.

*L994 caption Figure 7: Relationships (instead of: Dependence)*

AC: Caption modified

[revised manuscript text omitted]
 | 18 | $15.2 \pm 0.5$ | $36.05 \pm 0.13$ | $8.06 \pm 0.03$ | $-3.6 \pm 8.4$ | $0.65 \pm 0.37$ | $0.96 \pm 1.01$ | $0.14 \pm 0.06$ |
| ST2 | 16 | $21.0 \pm 1.3$ | $36.11 \pm 0.11$ | $7.97 \pm 0.03$ | $-10.3 \pm 5.7$ | $0.18 \pm 0.14$ | $0.42 \pm 0.60$ | $0.12 \pm 0.04$ |
| ST3 | 17 | $21.6 \pm 0.7$ | $36.09 \pm 0.28$ | $7.97 \pm 0.06$ | $-4.6 \pm 3.2$ | $0.24 \pm 0.29$ | $0.34 \pm 0.27$ | $0.09 \pm 0.03$ |
| ST4 | 17 | $17.7 \pm 0.7$ | $36.03 \pm 0.13$ | $8.05 \pm 0.05$ | $7.7 \pm 2.1$ | $0.46 \pm 0.33$ | $1.05 \pm 1.96$ | $0.23 \pm 0.09$ |
| ST5 | 16 | $15.4 \pm 0.3$ | $36.03 \pm 0.13$ | $8.09 \pm 0.12$ | $-19.1 \pm 9.4$ | $0.76 \pm 0.55$ | $0.68 \pm 1.17$ | $0.17 \pm 0.09$ |
| ST6 | 16 | $21.1 \pm 1.0$ | $36.37 \pm 0.05$ | $8.01 \pm 0.03$ | $-2.4 \pm 3.2$ | $0.26 \pm 0.34$ | $0.12 \pm 0.14$ | $0.10 \pm 0.05$ |
| ST7 | 17 | $20.6 \pm 1.2$ | $35.63 \pm 0.03$ | $7.94 \pm 0.03$ | $-2.6 \pm 5.0$ | $0.29 \pm 0.31$ | $0.37 \pm 0.50$ | $0.50 \pm 0.55$ |
| ST8 | 6 | $16.8 \pm 0.3$ | $36.44 \pm 0.04$ | $8.09 \pm 0.05$ | $-5.1 \pm 3.1$ | $0.69 \pm 0.32$ | $0.41 \pm 0.31$ | $0.14 \pm 0.11$ |

 *González-García et al., (2018).*González-García et al. (2018).

[revised manuscript text omitted]